# Concept and location neurons in the human brain provide the 'what' and 'where' in memory formation

Sina Mackay [1], Thomas P. Reber [1,2], Marcel Bausch [1], Jan Boström[3], Christian E. Elger [1] & Florian Mormann [1] ✉

Our brains create new memories by capturing the 'who/what', 'where' and 'when' of everyday experiences. On a neuronal level, mechanisms facilitating a successful transfer into episodic memory are still unclear. We investigated this by measuring single neuron activity in the human medial temporal lobe during encoding of item-location associations. While previous research has found predictive effects in population activity in human MTL structures, we could attribute such effects to two specialized sub-groups of neurons: concept cells in the hippocampus, amygdala and entorhinal cortex (EC), and a second group of parahippocampal location-selective neurons. In both item- and location-selective populations, firing rates were significantly higher during successfully encoded trials. These findings are in line with theories of hippocampal indexing, since selective index neurons may act as pointers to neocortical representations. Overall, activation of distinct populations of neurons could directly support the connection of the 'what' and 'where' of episodic memory.

The human medial temporal lobe (MTL) plays an essential role in memory. While many aspects of successful encoding and retrieval of mnemonic information have been extensively studied, the neuronal mechanisms that transform our perceptions into memories are as of yet mostly unknown. The main streams of information our brains need to access and combine in order to form new episodic memories are related to the question of "what" happened "where" and "when"[1]. A plethora of studies in rodents, non-human primates and humans have provided evidence for all three of these representations in the MTL[2–6].

The rodent literature has revealed different types of spatial representations such as hippocampal place cells[7,8] and entorhinal grid cells[2,9]. There is also evidence of neurons in the rodent MTL that are modulated by the temporal sequence of task events[10,11] or interactions of space and elapsed time[12]. Buzsáki and Tingley proposed a model of hippocampal function that assumes a less domain-specific organization of information, by relying mainly on sequences of relevant events[13]. The strong parallels between place and time cells[10] are in line with this notion. Some sequential[14] and temporal representations

including ramping cells[6] have also been shown to be reflected in neuronal firing patterns in the human MTL.

Spatial tuning in the form of grid cells[9] mapping two-dimensional space on a screen has been described in the entorhinal cortex (EC) of non-human primates[15] and has been linked to attention[16]. In humans, functional magnetic resonance imaging (fMRI) and magnetoencephalography (MEG) studies during virtual navigation have likewise provided evidence for hexagonal grid representations within the EC[17–19]. Another study described entorhinal cells tuned to upcoming target locations along a virtual track in humans[20]. Nevertheless, human entorhinal neurons have been shown to generally not be involved in the processing of scenes and spatial information[21].

Parahippocampal activity, on the other hand, has been linked to spatial navigation in 3D tasks on a laptop[22]. There is furthermore evidence of an allocentric coordinate system in the hippocampus of the moving macaque[23] and of both allocentric and egocentric representations in the human parahippocampal cortex (PHC) and hippocampus[24]. According to Bicanski and Burgess' elaborate model of memory and navigation, allocentric maps are computed in the hippocampus with the

[1]Department of Epileptology, University Hospital Bonn, Bonn, Germany. [2]Faculty of Psychology, UniDistance Suisse, Brig, Switzerland. [3]Department of Neurosurgery, University Hospital Bonn, Bonn, Germany. ✉e-mail: florian.mormann@ukbonn.de

help of bottom-up input from highly processed parietal sensory inputs. The system is alternating between a bottom-up and a top-down state so that mental maps can guide perception and can also constantly be updated with reference to moving objects or extended exploration[25]. Mediotemporal location- and view-specific neurons have been described in the human MTL ([26], but see ref. [27]), and grid-cell-like neuronal activity in spatial navigation has likewise been reported in the human EC and hippocampus[28,29].

A striking finding regarding selective hippocampal representations were visually selective neurons that represent semantic concepts[30] in different MTL regions including the hippocampus, amygdala, and EC. These neurons respond to the semantic content of a presented object or stimulus, e.g., to animals[31], pieces of clothing[32], or different pictures of a familiar person as well as to their written and spoken name[33]. They reflect subjective, conscious perception[34–36], and can be activated in the absence of stimuli during imagery[37], free recall[38], or mental comparisons referencing their preferred concept[39]. These neurons were named concept cells and have been hypothesized to represent the semantic building blocks of episodic memory[40]. The human PHC differs from the other three MTL regions by showing earlier and less selective responses[41] and no invariance to written and spoken words[33], by responding to scenes and spatial features of a stimulus[21], and by being involved in spatial tasks[42–45].

Previous studies have already addressed certain aspects of selective MTL activity in the context of memory tasks. These have yielded somewhat inconsistent results, such as significant modulation of firing rates during retrieval[46–48], but no effects on firing rates during encoding[48–50] with the exception of a population of egocentric spatial cells[24].

As pointed out by Wixted et al., one reason why it can be difficult to detect memory effects is that within a sparse coding system, those effects may only be exhibited by a small number of neurons[51].

In this study, we wanted to assess subsequent memory effects within the sub-population of visually selective neurons. We analyzed neuronal activity during the encoding trials of an associative memory task with moderate difficulty, allowing us to compare subsequently remembered to forgotten trials. Since there was a spatial component to our memory task we were also able to search for spatially tuned neurons and their response modulation with respect to memory formation. Given that we were able to pre-select items, but not locations, based on a preceding screening session (see Methods), we expected a larger number of item responses than location responses.

## Results

### Effects of experimental design parameters

Our associative memory paradigm involved sets of images presented at different locations within a 3 × 3 grid. The item-location associations

had to be recalled later upon presentation of the image beneath an empty grid. Subjects had to tap the location on the grid where the image had been presented during encoding. We recorded data from 3681 single and multi-units in 13 neurosurgical patients with bilaterally implanted depth electrodes in the amygdala, hippocampus, EC and PHC (Table S1). Stimuli were identified in a preceding screening procedure as likely response-eliciting (Methods). The task consisted of separate short runs where random combinations of images and spatial positions in a 3 × 3 grid on the screen had to be learned and then retrieved after a short distraction task (Fig. 1). Two variables were continuously adjusted in real-time during the task to achieve a performance of ~50%: the presentation duration during encoding, and the set size, i.e., the number of images that had to be memorized at once. The former could change after every trial, the set size had a greater impact on difficulty and was only adjusted after 3 consecutive high or low-performance trials (Methods). The presentation duration was modified in steps of 500 ms and held between 1.5 s and 3.5 s. Within this time window, the subject was required to tap the item location on the screen to verify that they had seen it. This triggered a green confirmation frame around the image but did not affect the presentation duration. Therefore, in all valid trials the reaction time (stimulus onset until confirmation tap) did not exceed the trial duration. We used linear mixed-effects models (Methods) to investigate the relationship between subsequent memory performance and the experimental parameters set size, reaction time and trial duration (in seconds). Our results revealed a significant effect of subsequent memory on set size ($\beta = -0.27$, $P < 10^{-16}$, subsequently forgotten trials were in larger sets) and on reaction time ($\beta = 0.02$, $P = 0.01$, subsequently forgotten trials had longer reaction times), but not on trial duration ($\beta = 0.004$, $P = 0.9$).

### Subsequent memory effects in selective neurons

Due to the preceding screening procedure, we found highly significant fractions of neurons responding selectively to one or more items in all recorded brain regions (all $P < 10^{-43}$, binomial test, one-sided with $n =$ total number of neurons per region, $k =$ number of responsive neurons per region, $P = 0.001$, corresponding to the alpha level of our response criterion, refer to "item responses", Fig. 2, Fig. S1A for examples). In addition to the binomial test, we calculated the empirical size (i.e., the probability of falsely rejecting the null hypothesis if it is true) in each measured brain region. To this end, we compared the fraction of responsive items to 10,000 realizations of label-shuffled data and found empirical sizes of $alpha < 10^{-4}$ in all measured brain regions (Fig. S6). In this case, the binomial test (nominal size) and label-shuffling test (empirical size) produced consistent results. Responses to items were detected using a binwise rank-sum test ($P < 0.001$, see Methods). Whenever a neuron responded to one or

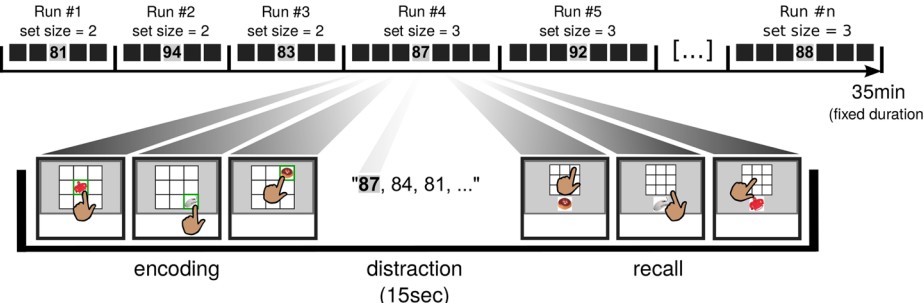

**Fig. 1 | Experimental design.** Top row: Each experimental session had a fixed duration of 35 min and consisted of consecutive runs of varying content and number of encoding trials, dynamically adapted to the subject's performance. Bottom row: Composition of a single run. Each run consisted of encoding trials, a distractor task and retrieval trials. In this example, all locations were remembered correctly. Images used in this figure are licensed. Copyright © 2001 Thomas Reber and Getty Images. All rights reserved.

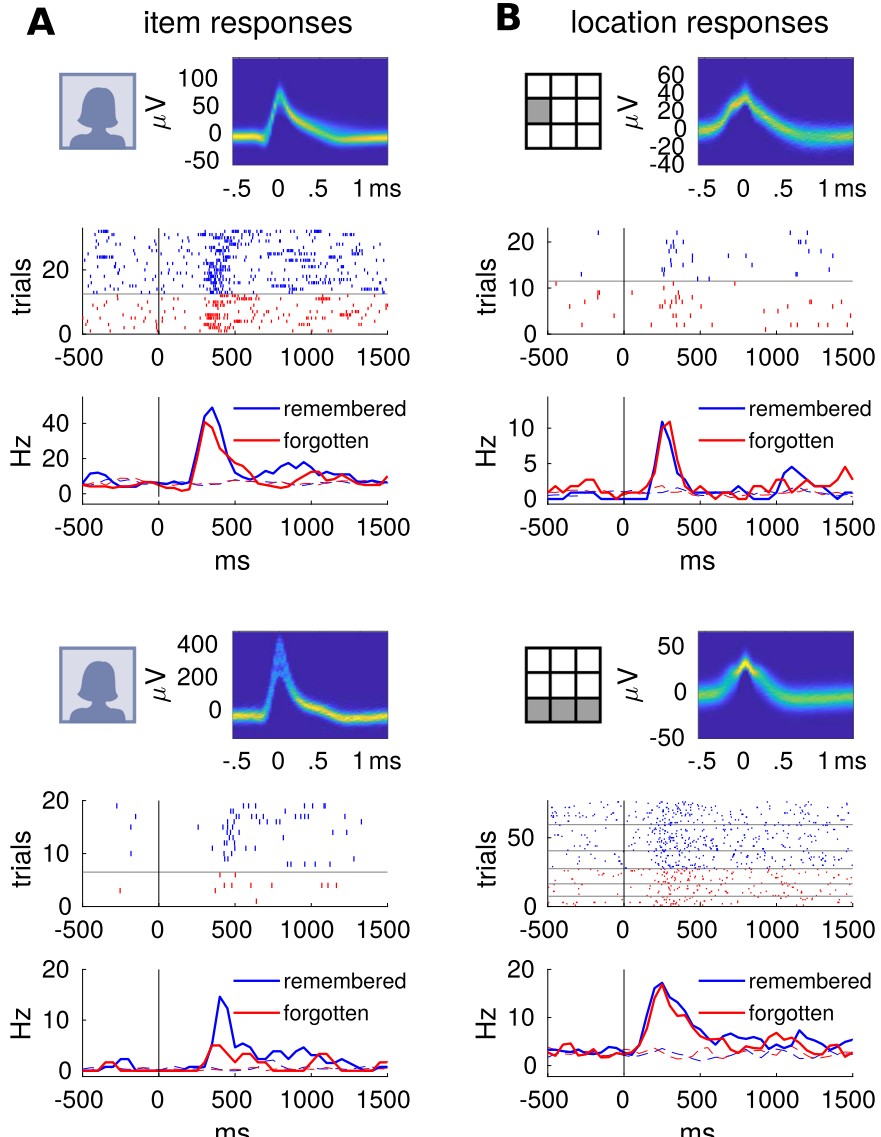

**Fig. 2 | Examples of item and location-specific responses. A** Selective responses by single neurons (top: amygdala, bottom: hippocampus), separated based on correct vs. incorrect subsequent retrieval. Solid lines (lower panels): response to the preferred item. Dashed lines: average response to all non-preferred items (cf. Fig. S1A). **B** Responses of single neurons in the PHC to spatial locations within the presentation grid. Solid lines (lower panels): response to the preferred item locations, which in the lower example includes the entire bottom row of the grid.

Dashed lines: average response to all non-preferred locations (cf. Fig. S1B) Subsequent memory effects per neuron were statistically assessed using a one-sided Wilcoxon rank-sum test for the time window of 0 to 1500 ms. Statistically significant effects were found for the two item neurons (top, $P = 0.008$, $Z = 2.40$; bottom, $P = 0.02$, $Z = 2.04$), but not for the two location neurons (both $P > 0.1$, $Z < 0.95$). Source data are provided in a git repository (see Data Availablity).

more items, we computed the response activity for this neuron by averaging all trials containing a preferred item.

Responses to grid locations were computed in the same way in that a neuron had to show a significant response to one or more of the nine locations in which an image was presented throughout the experiment. We furthermore found the responsive cells to be selective. The vast majority of these neurons responded to half or fewer of the presented items (Amygdala: 99%, Hippocampus: 96%, EC: 100%, PHC: 84%), or item locations (PHC: 81%, see below). There were two hippocampal target locations (anterior and posterior hippocampus), which were grouped together in all analyses. The fractions of responsive neurons in these two hippocampal regions did not differ within patients (item neurons: $T(12) = -0.64$, $P = 0.54$, location neurons: $T(12) = -1.53$, $P = 0.15$, paired *t*-test). Figure 3 shows that item responses were modulated by subsequent memory performance in the

amygdala, hippocampus, and EC, in that the responses to subsequently correctly placed items were more pronounced. This effect occurred at a latency of 239 to 1249 ms in the amygdala, 531 to 796 ms in the hippocampus, and 491 to 618 ms in the EC, i.e., generally after the initial peak activity (250 to 500 ms, see also ref. [41]). Notably, this effect was not observed for item responses in the PHC.

We also investigated responses to spatial locations, i.e., to squares within the presentation grid. Using a binomial test, we found significant fractions of neurons responding to locations in the amygdala, hippocampus, and PHC (all $P < 10^{-23}$, one-sided with $n$ = total number of neurons per region, $k$ = number of responsive neurons per region, $P = 0.001$, corresponding to the alpha level of our response criterion). However, it is important to note that the nominal size (significance level) might not always align with the empirical size of the test. Specifically, to test whether more than 0.001 of cells could be expected to

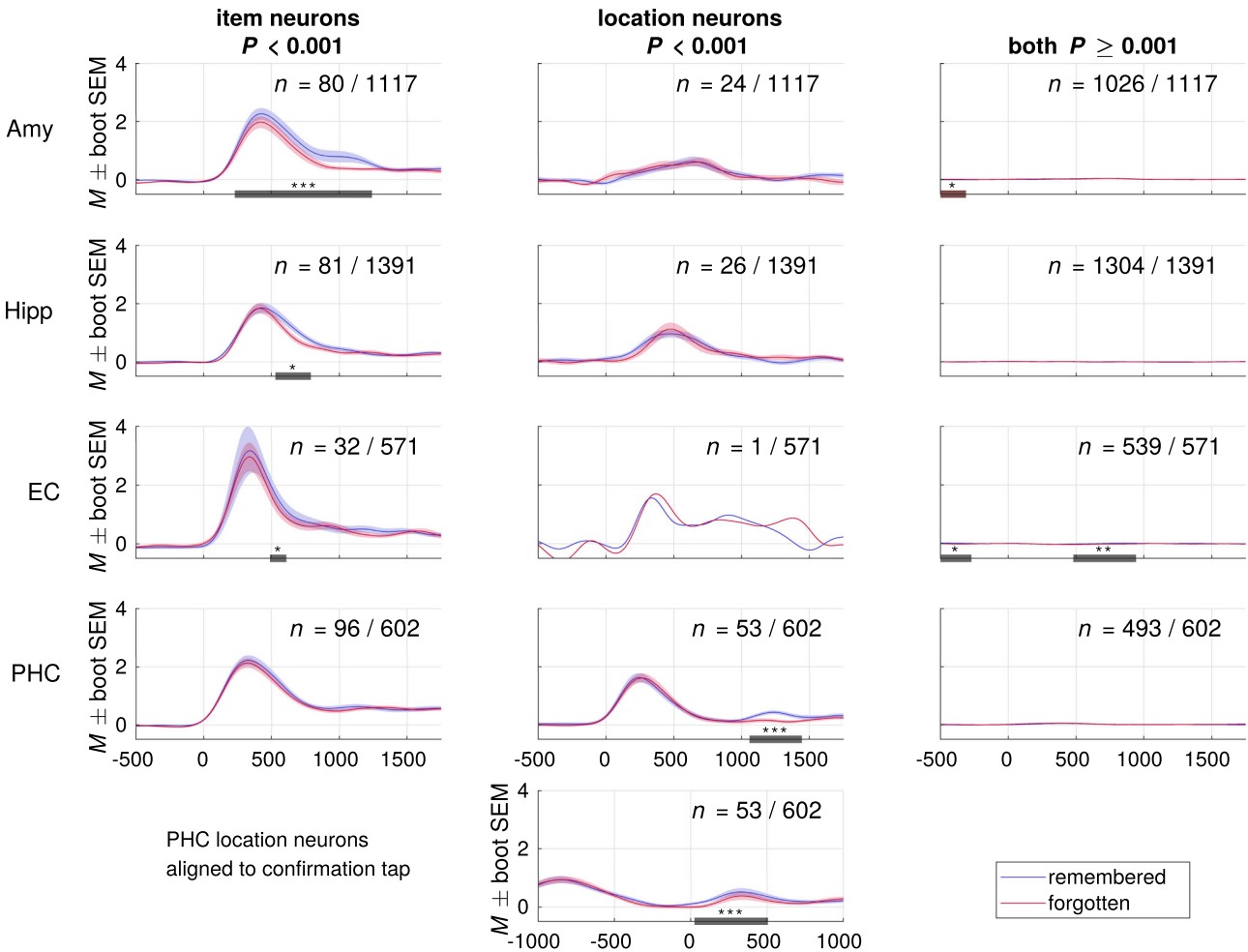

**Fig. 3 | Item and location-specific group responses predict subsequent memory performance.** Group activity as averaged convolved firing rates during responses in encoding trials, aligned to stimulus onset. For neurons responding to several items or locations, all trials featuring a response-eliciting item or location were averaged. Shaded areas denote standard errors of bootstrapped means. Subsequently remembered (blue) and forgotten (red) trials were compared using a cluster permutation test (indicated as dark bars along x-axis, *P < .05, **P < .01, ***P < .001, see Methods). The n indicated in each panel is the number of neurons fulfilling the (non-) response criterion. A responsive neuron can be represented in the left and middle column, so that the n summed across one row may exceed the total population. Each row represents one of the four recorded brain regions. Left column: all neurons with a significant response to at least one item. Maximum effect sizes and cluster P values: Amygdala (Amy) $d = 0.36, P < 10^{-4}$, Hippocampus (Hipp) $d = 0.33, P = 0.010$ and EC $d = 0.21, P = 0.038$. Center column: all neurons with a significant response to at least one spatial location. Effect size in PHC: $d = 0.60, P < 10^{-4}$ (aligned to stimulus onset), $d = 0.37, P < 10^{-4}$ (aligned to confirmation tap). Right column: all remaining neurons. Effect size in Amygdala: $d = -0.10, P = 0.046$ (pre stimulus onset), EC: $d = 0.23, P = 0.031$ (pre stimulus onset) and $d = 0.14, P = 0.005$ (post stimulus onset). The right column is also displayed in Fig. S3, with adjusted y-axis ranges. Source data are provided in a git repository (see Data Availablity).

be responsive by chance, we compared the measured fractions to 10,000 iterations of label-shuffled data. We only found a significant empirical size for the PHC (0.0027), but not amygdala (0.23) or hippocampus (0.81, Fig. S6). We then statistically compared the proportion of location cells found in PHC to that in all other regions. Out of all parahippocampal neurons, 8.80% responded to at least one of the squares in the presentation grid ("location responses", Fig. 2, Fig. S2), a significantly higher percentage than in the amygdala or hippocampus (chi-square test, both $P < 10^{-8}$, $\chi^2 = 40.50$ and 53.08, Fig. S2). These location cells also showed a subsequent memory effect. As with the item responses described above, firing rates were higher in subsequently remembered trials. This effect was found in a later time window (1059 to 1444 ms), subsequent to or partially overlapping with the effects in item responses in the amygdala, hippocampus and EC. Since this effect overlaps with the median response latency of 1.16 s (image onset until confirmation tap), we also evaluated the same responses aligned to the response tap (Fig. 3, bottom panel). The reactivation did not seem to be driven by motor processes since it took place after the

tap (25 to 506 ms) and it was significantly modulated by subsequent memory performance for both alignments.

In the amygdala and EC, we saw a subsequent memory effect in neurons exhibiting no significant item or location responses. The effect sizes of $d = 0.10$ (Amygdala) and $d = 0.14$ (EC), however, were considerably smaller than that of the memory effects previously described for item and location responses, which ranged from 0.21 to 0.59 (Fig. 3). Since the two traces are difficult to discern in column 3 of Fig. 3, refer to Fig. S3 for the same plots with an adjusted y-axis.

## Neural activity during delay periods
Between the encoding and retrieval trials of each run, we prevented any rehearsal strategies by adding a 15-s counting task (see Methods). We were nevertheless interested in whether neurons were reactivated during this delay period. For each neuron responding to exactly one stimulus, we computed the average firing rate normalized to the 500 ms preceding stimulus onsets in encoding trials. Two averages were calculated for each neuron, one across counting episodes during

which the preferred item's location was remembered and one for episodes during which it was forgotten. We then compared those two values across neurons using a paired *t*-test and found no significant differences ($T(191) = -0.68$, $P = 0.50$). This analysis was repeated for each brain region (all $P > 0.1$) and also for location-selective neurons in the PHC ($T(26) = -0.21$, $P = 0.84$).

## Control analyses

Since a small fraction of neurons were classified as both item and location neurons, we repeated the main analyses shown in Fig. 3 after excluding these neurons and found the effects to be largely identical (Fig. S4).

Given that images were shown repeatedly across trials, we tested for effects of adaptation or memory interference from previous trials by performing a split-half analysis (first half of trials vs. second half of trials). Both halves showed quantitatively similar results to those shown in Fig. 3 (data not shown here).

Furthermore, we verified whether the preferred stimuli of selected cells remained the same during the retrieval trials. Indeed, firing rates in response to preferred items were higher in all four recorded brain regions (all $P < 0.001$ for amygdala, hippocampus, EC, and PHC, two-tailed signrank test, see Fig. S5) during retrieval. The same was true for location cells in the PHC ($P < 0.001$), supporting the idea that location cells are in fact encoding location and not merely combinations or associations. For these tests, normalized mean firing rates were computed during the 1000 ms leading up to the response tap in retrieval trials, and compared using signed-rank tests (Fig. S5).

## Discussion

Our mnemonic recall of an experienced event or episode comprises among other things the information of where the event or episode happened, who or what was involved, and when it took place. The MTL's task during the encoding of such an episodic memory thus consists of associating corresponding representations at the neuronal level. In this study, we operationalized this association of "where" and "what/who", i.e., of item and location information, in the form of an associative memory task. The adaptive design increased difficulty to the point where the participants were not able to remember all information. Furthermore, every delay period was filled with a 15 s backwards counting task. This resulted in a high memory load and prevented the rehearsal of the learned associations. Based on these features, the completion of this task requires long-term memory[52,53].

Two independent studies have reported large fractions of visually selective neurons in the amygdala, hippocampus and EC to exhibit invariance with respect to different visual representations of the same semantic concept[33,54] (72% across amygdala, hippocampus, and EC, and 77% across amygdala and hippocampus, respectively), A previous study from our own group required a high level of abstraction for neurons to qualify as concept cells[39], which, again, was the case for the majority (53%) of neurons across the same three regions. We therefore expect a majority of item neurons to qualify as concept cells.

Previous studies investigating memory encoding in the human MTL at the neuronal level have shown subsequent memory effects only at the population-code level. These studies find that the majority of memory-predictive cells show increased firing rates during encoding when information was processed that could later be recalled[55] or recognized[56]. However, these effects have largely been absent in selectively responsive single neurons ([6,48–50] but see ref. 24). Our results are in line with the idea that concept cells represent the building blocks of memory[40]. Not only do we see subsequent memory effects; they are also restricted to sparse, selective neurons[51]. In the amygdala, hippocampus, and EC this applies exclusively to item neurons, suggesting that they provide the "who/what" information in associative memory encoding. Following this theory and considering the analogous effects with regard to "where" representations in parahippocampal location

neurons, this population could provide spatial information for memory encoding. The PHC being home to location neurons is in agreement with a number of other studies tying parahippocampal activity to spatial tasks[42–45]. Another property of the PHC that is consistent with earlier findings is its lower degree of selectivity[21,41], which we see in responses both to locations and to items (Fig. S2).

The hippocampal memory indexing theory[57,58] offers an interesting framework with respect to our findings. This theory's core idea is that in order to encode an event, a hippocampal code, or "index", is created which points to neocortical networks where information associated with the event is stored[58]. Through coordinated activation of index neurons for different concepts, synaptic connections between different index neurons or between their respective referenced neocortical networks could be strengthened via spike-timing-dependent plasticity[59]. A thorough and extended activation of index neurons representing concepts could thus facilitate a connection to neurons representing a spatial location. In light of our data, we see the memory-predictive item neurons in the hippocampus, and also the amygdala and EC, as potential pointers to neocortical semantic content. They could fulfill the role of the "index" according to the hippocampal indexing theory and support memory encoding. Refs. 13, 60 posit that these types of pointers should be content-free and part of pre-defined sequences that can be assigned as needed to contents such as experienced events. We routinely identify concept cells in screening sessions to investigate them in follow-up experiments later during the day and find their responses to the same stimuli to be trackable for hours or even days using standard monotrode microwire recordings. These concept neurons, therefore, appear to be permanently and invariantly (i.e., independently of context) assigned to a semantic content and not easily re-assignable to new perceived concepts on the fly (but see ref. 61). This observation of invariance over time ties in with the general idea that the human memory system might be optimized for creating semantic associations rather than ordered sequences.

It is worth noting that location neurons in the PHC showed a subsequent memory effect, but that there was no corresponding effect in the respective item neurons in this brain region (Fig. 3). Since the fractions of location cells in the amygdala and hippocampus were not statistically significant in the label-shuffled permutation test, any response activity to a specific grid location in these regions (Fig. 3) is likely an epiphenomenon of response activity to the visual stimuli. We only found a significantly large population of location cells in the PHC, which was also the only brain region to respond more strongly to the same preferred locations during retrieval as during encoding (Fig. S5). Together with the aforementioned distinctive features of the PHC, this could indicate that the parahippocampal location neurons are not pointers, but actual neocortical representations based on a population code[21].

The firing behavior of entorhinal neurons was of special interest since this region is closely linked to both PHC and hippocampus. In this experiment, we observed firing behavior in the EC to resemble that in the amygdala and hippocampus, rather than the PHC. Some previous findings point towards entorhinal involvement in spatial navigation[20,24,48], yet we found hardly any responses to spatial locations. One possible explanation for this discrepancy is the lack of egocentric navigation required in our task. The layout on the screen is more reminiscent of a map, which is rather linked to semantic knowledge[62]. In another study, entorhinal neurons did not show the same strong preference for landscapes as the PHC[21], which is in line with our results.

The finding that the subsequent memory effect in location responses occurred in a later time window than that for item responses could result from the way in which humans process "what" and "where" information. There are several linguistic models of thematic hierarchy which differ slightly depending on the phenomena they aim to explain. They rank semantic elements of sentences such as the agents, experiencers,

goals, location, instruments, etc. according to their prominence. Almost all of them rank location in the lowest category[63]. Furthermore, there is evidence of a universal, natural order in which humans convey information when forced to use gestures instead of the spoken language they are used to. In a study where scenes with one stationary and one moving object were watched and then reproduced, the objects were acted out before the spatial movement[64]. Perhaps the order of the effects we see on a neuronal level, namely item before location, reflects the architecture of internally generated narratives, where information components are processed in descending order of prominence.

A third stream of information that has been suggested to be integrated in the process is temporal, i.e., the aspect of "when" something happened[1]. While some researchers have described neuronal activity related to passing time[6], it is difficult to assess temporal activity entirely independently of other relevant aspects of the experimental task or the subject's behavior[12,13,60].

Episodic and semantic memory are the two constituents of declarative memory, which, unlike implicit memory, requires explicit conscious perception of sensory input. The activity of concept neurons in hippocampus, amygdala, and EC indeed has been shown to follow conscious perception rather than stimulus input[36].

Note that due to its experimental task implementation, our study was not designed to investigate memory consolidation, a process during which memory traces are stabilized and presumably transferred to the neocortex to eventually become hippocampus-independent. Instead, we deliberately prevented active rehearsal between the encoding and retrieval phase by means of our mathematical distraction task. It can be hypothesized, however, that mediotemporal concept neurons and possibly also parahippocampal location cells involved in our everyday experiences are reactivated during periods of memory consolidation, e.g., during slow-wave sleep[65]. Such a reactivation of pointer neurons during an offline consolidation state with no sensory input could likewise facilitate the strengthening of synaptic connections between the neocortical representations referenced by mediotemporal pointer cells. Future studies will be needed to investigate this hypothesis.

## Methods

### Participants and setting
We recorded data from 13 in-patients (20–62 years old, 8 female, 5 male) with drug-resistant epilepsy who had undergone invasive surgery for seizure localization. Due to the implanted electrodes that were wired to the recording system, the patients were confined to their beds for around 7–10 days. During this time, we ran our experiments with them in their hospital beds. They sat up at least 45° and performed the task on a touch-screen laptop on a tray in front of them. All participants gave their written informed consent, and the study was approved by the Medical Institutional Review Board of the University of Bonn.

### Screening procedure
Each recording was preceded by a screening session in the morning of the same day in order to identify response-eliciting images. This screening session was either an object screening (OS) with a fixed set of 100 images of commonly known objects and animals described in a previous publication[32], or a customized person screening (PS) with an individual set of 100–150 images of the participant's friends and family, public figures, familiar places or objects related to their hobbies and jobs. These screenings were very similar in experimental design to the procedures described in previous publications from our own and other groups[30,31,33,35,41,66]. Each image was shown 10 (OS) or 6 (PS) times and a simple decision task was performed after every presentation (OS: "Is the object man-made?", PS: "Does the image contain a face?"). The repeated presentation of each image allowed for the detection of statistically significant responses to certain images. The images shown during the screening covered a large number of semantic concepts,

and the stimuli selected for our main task generally depicted different objects, places or people.

### Task
The spatial framework of the main experiment was a $3 \times 3$ grid on a touchscreen laptop, and each image was presented in one of the 9 squares. The task was to remember and retrieve the spatial locations of the images. Each session was limited to 35 min and was divided into runs (Fig. 1, top row), where the total number of runs varied depending on the speed and performance of the patient. Within each run, a subset of images was shown, one at a time, at different, randomly assigned locations within the grid (Fig. 1, bottom row). The participant was asked to confirm every image location by tapping it within the presentation time window (1.5–3.5 s). Whenever the correct square was tapped, a green square appeared along its outline for the remainder of the presentation duration. Trials with off-target or missing confirmation tap were considered invalid and were excluded from the analyses assessing memory effects. Those trials also triggered an immediate dissonant feedback sound and, in case of a misplaced tap, a red square around the tapped, empty square. Following the encoding trials, a random number between 80 and 100 appeared on the screen and the participant counted down vocally in steps of three until the number disappeared after 15 s. The last part of each run consisted of recall trials, where one by one the items from the beginning of the run were shown in shuffled order below an empty grid and the participant tried to recall and tap each item location. After each run, a feedback screen showed the percentage of correct answers. Retrieval trials and initiation of runs were self-paced. For each new run, a new subset of images was drawn from the item pool, evening out presentation counts, and filling remaining slots by random selection. The locations were assigned randomly. In order to obtain similar numbers of subsequently remembered and forgotten trials, we adjusted the difficulty in two ways. Each run was classified either as high-performance (>65% correct), low-performance (<35% correct) or medium performance (35–65% correct). Encoding presentation duration was initially 2 s and was increased following low-performance trials and decreased following high-performance trials. Values changed in steps of 0.5 s and were capped at 1.5 s and 3.5 s. Furthermore, after 3 consecutive low-performance trials of equal set size, the set size of the next run was decreased by 1 and accordingly increased by 1 after 3 consecutive high-performance trials. Whenever the set size changed, presentation duration was reset to 2 s. The minimum number of images per run was 1, the theoretical maximum was the total item pool size for the session (up to 8, details below), which was only reached in one session but was not a limiting factor. The initial set size was always 2, ensuring a low difficulty and therefore high motivation for most participants. This resulted in relatively high performance during the initial runs, and overall we recorded more correct than incorrect trials (13.3 vs. 10.7 on average).

The item pool size for an entire recording session was between 4 and 8 and was based on expected patient performance. Low memory performance would result in smaller set sizes and thus lower overall trial counts within the time limit of 35 min. Aiming for similar numbers of presentations per image across patients, we determined smaller image pools for putative low-performance participants. The mean number of trials per session was 168.61 (sd 49.88, range 64–278), the number of runs was 58.83 (sd 12.30, range 34–83), and the mean set size was 3.18 (sd 1.17, range 1–7).

We did not expect epilepsy-related neuronal firing to substantially affect our results. As shown by ref. 67, such interference should be minimal and should only affect small, specific sub-populations during recall.

### Electrophysiological recordings
All data presented here were recorded from implanted Behnke-Fried depth electrodes (AdTech, Racine, WI), inserted through the hollow

clinical macro electrodes, and protruding from the tips by ~4 mm. The microelectrodes were grouped in bundles of 8 recording wires plus one reference wire per macro electrode. The standard bilateral implantation scheme included 5 bundles per hemisphere, 1 in amygdala, 2 in hippocampus, 1 in EC, and 1 in PHC, adding up to 80 recording microwires in total. The continuous signal was recorded at 32 kHz on a Neuralynx ATLAS system (Bozeman, MT). Spikes were extracted and semiautomatically sorted using Combinato[68]. This software includes several mechanisms to automatically detect and reject artifacts: removal of spikes during extremely high firing rates, high amplitude events, overlapping spikes, and events detected concurrently on many channels. Automatically pre-sorted units were manually verified, adjusted where necessary, and classified as single units (SU), multi-units (MU), or artifacts based on spike shape and variance, signal-to-noise ratio (SNR), the inter-spike interval distribution of each cluster, and presence of a refractory period for the single units. We calculated the SNR for each single and multi-unit. It was defined as the mean spike amplitude divided by the median absolute signal. Single units (median SNR 2.85) had significantly greater SNRs than multi-units (median SNR 2.08, $P < 10^{-38}$, rank-sum test). We recorded a total of 3681 neurons (1816 single units and 1865 multi-units) in 44 sessions, specifically 1117 units from amygdala, 1391 from hippocampus, 571 from EC and 602 from PHC.

### Responsiveness and statistical tests

To determine responsiveness, we used an established criterion based on a binwise rank-sum test (100 ms windows, 50% overlap, 0–1000 ms post stimulus presentation) with Simes correction for 19 bins[41] and a significance level of $alpha = 0.001$. Whenever responses were compared with regard to subsequent memory, we calculated one average response per neuron across respective trials. In the case of several response-eliciting items, all trials depicting any of those items were averaged. The same applied to location responses.

The number of responsive neurons was then tested against chance levels for each brain region, using two different approaches. One was a parametric approach, a binomial test where the occurrence rate $P$ was set to $P = 0.001$, the same as the alpha level in the response criterion described above. The other was a permutation-based approach, where item labels or location labels were shuffled 10,000 times, resulting in a distribution of 10,000 proportions. The $P$-value was calculated as the fraction of label-shuffled data points that were more extreme than the measured data including half of shuffled data points that were equal to the measured value.

The population responses in Fig. 3 were then compared using a cluster permutation test[69]: first the responses during subsequently remembered and forgotten trials were compared at every time point, resulting in temporal clusters of significant differences (paired $t$-test $P < 0.05$) between the two conditions. The same was done in 10,000 iterations of label-shuffled data. Finally, the cluster sizes from the true data were ranked against the distribution of cluster sizes from the shuffled data. Only clusters whose size ranked in the top 5% were considered and marked by the dark horizontal lines in Fig. 3.

### Linear mixed-effect models

We investigated whether there was a relationship between subsequent memory performance and the three experimental parameters set size, trial duration, and reaction time. To this end, we first calculated two means for each of these parameters per session, one across all subsequently remembered encoding trials and one across all subsequently forgotten encoding trials. Accordingly, subsequent memory performance was used as the predictor to model the relationship. To account for individual differences between patients and across sessions, we fitted linear mixed-effects models with random intercepts

and slopes at the level of patient ID and session index (nested within patient ID). The reported estimates $\beta$ in the Results section refer to the fixed slopes (i.e., average slopes across all patients).

### Reporting summary

Further information on research design is available in the Nature Portfolio Reporting Summary linked to this article.

## Data availability

The data used to produce all figures is available in the git repository detailed below, within the directory "source_data". Source data are provided with this paper.

## Code availability

The code for producing all figures is available along with the source data, in the git repository https://github.com/s-mackay/grid_memory.

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

## Acknowledgements

We thank all patients for their participation, Johannes Niediek for discussion, and Gert Dehnen for technical assistance. This research was supported by the Volkswagen Foundation (86 506), the German Ministry of Education and Research (BMBF 031L0197B), the German Research Council (MO930/4-2, MO930/15-1, SPP 2205, SPP 2411, SFB 1089) and a NRW Network Grant (iBehave).

## Author contributions

Conceptualization: T.P.R., F.M., Data acquisition: S.M., T.P.R., M.B., Analyses: S.M., Methodology: S.M., F.M., Patient recruitment: C.E.E., F.M., Neurosurgical procedures: J.B., F.M., Funding acquisition: F.M., Project administration: F.M., Supervision: F.M., Writing – original draft: S.M., F.M., Writing – review & editing: S.M., F.M., T.P.R., M.B., C.E.E., J.B.

## Funding

## Competing interests

The authors declare no competing interests.
