## [Peer Review File · Nature Communications]

REVIEWER COMMENTS

Reviewer #1 (Remarks to the Author):

Summary: The authors examine pre-selected neuronal activity from thousands of neurons from 15 patients (across 44 sessions) with medically refractory epilepsy during a memory task that is oriented towards locations and identities of objects that neurons were selective for. The encoding and recall portions of the task were segregated with a 'math distractor' task in which patients were instructed to count backwards from a random number for 15 seconds. The authors discovered that substantial proportions of amygdala and hippocampal neurons were selective for remembered vs. forgotten objects. This study is interesting and has cool, invaluable human neuron data and would be appropriate for publication after minor changes and added controls.

Minor critiques:

- PHC isn't abbreviated in the intro until after the authors already use the full word several times.
- A t-test is inappropriate for the behavioral data shown for the hierarchical data in Figure 1B. A mixed effects model should be used with patient as a random factor to control for any variance contributed across subjects.
- Important to list how many of the unit clusters were isolated units, and how many were labeled multiunit.
- It would be nice to show some sort of unit isolation or signal-to-noise ratio metrics for the units.
- It looks like the EC neuron in Figure 3 has a response difference at around the same time as the PHC neurons that appears to be one of the larger effects reported in the paper. Is that difference due to outliers? Which factors did the authors correct for in their cluster based permutation tests?
- It is unclear what group of neurons (or trials?) the authors are referring to on page 7, line 177, "...the latter group of higher firing rates..."
- In the task section of the methods, the authors refer to 'figure 1, top row.' Do they mean 'Fig. 1A, top row?'
- The claim that the 15 s delay constitutes a "long term memory task" may require further justification.

Major critiques:

- Were there any differences in proportions of neurons encoding 'what'/'where' in anterior/posterior hippocampal neurons? It's unsurprising that the authors find fewer location-selective neurons, as the location of the item did not need to be remembered to complete the task.

- There is no mention of how epileptiform discharges were removed.
- The paper is light on controls. What sort of controls could be carried out in order to rule out alternatives for the claims of the paper? Did they have a control brain region at all? To see if neurons' firing rates outside of the MTL are also predictive of memory performance?
- The lack of EC neuron response is contrary to much of the literature that shows EC is necessary for spatial memory. The authors should discuss this discrepancy.

Elliot Smith (Neurosurgery) @ University of Utah

Reviewer #2 (Remarks to the Author):

In this study, the authors recorded 3681 single-units while epilepsy subjects performed an associative memory task, and demonstrate that neurons tuned to “concepts” and “location” both modulate their firing rate depending on whether the stimulus in question is remembered or not. To do so, they first identified image-selective neurons (inferring that many of these are “concept cells”) during a screening task. Then, during encoding, they identified location-selective neurons. Finally, they show that image-selective neurons in the Hipp, Amy, and EC differentiate between subsequently remembered and forgotten items, while location-selective neurons in the PHC do so. This is an interesting finding, but the key interpretations depend on analyses that require further clarification. These include (in no particular order):

1. The authors never describe how location-selective neurons are identified, simply stating that neurons “responded to at least one of the squares in the presentation grid”. What does this mean? I have to infer that the same bin-wise rank-sum test used for concept-cell identification during screening was used during stimulus presentation in a specific square during encoding. But the images are on screen for variable amounts of time – what window of time was used? Were responses between locations compared statistically (i.e. by using an ANOVA)? Was there some kind of cut-off for the # of locations that a neuron could respond to before being deemed non-selective for location?
2. While more straightforward, the authors do not provide any information on how many images neurons were selective for. This is important because several subsequent analyses depend only on trials including images that a neuron was selective for.
3. The authors state that “We found significant fractions of neurons responding to locations in the amygdala, hippocampus and PHC (all $p < 10^{-15}$, binomial test)”. However, I conducted a binomial test

myself in Python, where $p(\text{observing cell-counts by chance} \mid \alpha=0.05) = \text{scipy.stats.binom_test}(\text{cell-count}, \text{total cells}, 0.05)$, and found p-values of 0.001, 0.16, and 2×10^{-23} for amygdala, hippocampus, and PHC. Perhaps I made an error here – can the authors account for this discrepancy? This is relevant to #6, below in particular.

4. I am not fully convinced that the two primary statistical approaches here play nice together with respect to timing. First, the authors categorize neuronal selectively by binning time (100 ms bins, 50% overlap) and looking for at least one bin that shows significantly elevated firing above baseline, corrected for multiple comparisons. If a neuron shows elevated firing in even one time-bin it is considered to be selective for item or location. However, the specific timing of elevated firing does not matter during categorization – just that neurons exceed baseline in one or multiple time-bins. There could be great variability in the specific timepoints of high firing. Yet, the cluster-permutation test used for the SME is more like an ERP, smoothing over this variability by averaging over cells and identifying specific adjacent-timepoints that distinguish remembered and forgotten trials across cells. If specific timepoints only matter for the latter analysis, and not the former, a convincing control analysis would be to categorize cells as location or item selective based on mean firing rate during stimulus presentation, and not using bin-wise categorization. This might exclude neurons who show very brief increases in firing rate, and might include neurons with more diffusely elevated firing – do the cluster permutation results survive in this case? Relying on mean firing rates could also simplify the SME calculation, resulting in a single test between recalled and forgotten trials, as in PMID: 34742943.

5. Are there any neurons that were image-selective during screening and location-selective during encoding? If so, what is their proportion? This is crucial to understanding the specificity of these results.

6. Judging from Figure 3, there is a very small proportion location-responsive neurons in the Amy, Hipp, and EC, yet on Line 229 the authors conclude that location-selective cells only exhibit SMEs in PHC. A more appropriate conclusion would be that location-selective cells only exist in PHC. Can the authors justify drawing conclusions about location-selective SMEs from population-level firing-rate data in Amy, Hipp, and EC, when the data come from a population small enough that it could have been observed by statistical chance? This might be relevant to image-selective cells, too (see #3).

7. Are location cells really encoding location? How do the authors rule out that these neurons are conjunctively coding an association between items-locations, rather than location alone? One way to do so might be to measure if the same neurons fire when subjects click on the preferred locations during recall – independent of whether that recall was correct or not, the neuron should show some heightened activity in that location that is not dependent on the presence of a visual stimulus in the square.

8. Subsequent-memory effects (SMEs) in the image-selective neurons are only ascertained in trials with image-selective stimuli. What happens in trials without image-selective stimuli? Obviously the overall firing rates may be much, much lower, but is there still a difference between remembered and forgotten items? This is important for judging the selectivity of this SME for the content that is thought to be encoded by these neurons.

9. In order to assess how the 3,681 neurons were isolated across regions, subjects, and sessions, authors should include a demographic table for the 13 subjects which includes a count of sessions-per-subject, and # of isolated neurons isolated in each region, by session.

10. The authors refer to their image-selective neurons as “concept cells”, despite not actually testing them for concept-specificity. I understand this follows from PMID: 28218914, but the inference being made there (and here) is that some large subset of visually-responsive neurons in these regions are concept cells. This is contradicted by Figure 4 from the same lab’s prior work (PMID: 34697305), which shows that in many of these regions (specifically PHC), many cells are only visually-responsive, and not conceptually consistent. As I point out in #6, the proportions of image-selective cells is actually quite low outside of the PHC, and if only some subset of this already-small population is concept cells, then I hesitate to draw too many conclusions about “concept cells” from these results. I strongly suggest the authors rephrase this to “visually-selective” cells.

11. The authors should cite prior evidence in humans that location-sensitive neurons show SMEs (PMID: 31902728), as well as SMEs during free recall (PMID: 34742943).

12. The discussion of “hippocampal indexing theory” does not jive with the data as presented. I’m not positive there’s a significant # of image-selective cells in hippocampus (see point #3). I’m not sure these image-selective cells are “concept cells” (see #11). The authors do not provide a lot of support for their proposal of the PHC’s role, given the large number of image-selective cells there (with no SME) and location-selective cells (with SME).

13. The discussion of the relative timing of SME effects is also not convincing given that the peak firing rate responses of both image- and location- selective cells seems to be in the same range (200-700 ms), with the SME effects in the location-selective neurons clearly aligning to the time of the button-press.

Reviewer #3 (Remarks to the Author):

Mackay and colleagues investigated the contribution of visually-responsive “concept cells” to human associative memory using microwire recordings in epilepsy patients. Following a screening session to identify images for the memory experiment, patients learned and retrieved the locations of images in a 3x3 array on a computer screen. The authors found that concept cells differentiated their firing during memory encoding and also observed neurons modulated by the location of images in the array. Other neurons did not show subsequent memory effects. The findings are interpreted as evidence that concept cells support episodic memory formation. If true, these findings would be an important and timely contribution to our understanding of the electrophysiological basis of human memory. However, the manuscript is missing key analyses and lacks sufficient description of methods for me to be wholly convinced at this point. I am nonetheless hopeful that this manuscript and dataset could be published following major revisions.

Major strengths:

The authors utilize an impressive sample size of single- and multi-unit data to assess their question.

I applaud the authors for using an adaptive and elegant memory task which allows them to test memory in patients with varying cognitive function.

Major weaknesses:

This experiment critically relies on the screening session to identify stimuli for the memory experiment yet there is very little description of the screening task. Please add a full description of the screening session.

From what I can discern, only those stimuli eliciting responses in the screening session were included in the memory experiment. If true, I believe this may potentially lead to circularity in the authors conclusions because the authors did not include images that did not elicit responses as an experimental control.

Images were re-used across runs in a session. While the authors audibly included a “confirmation tap” in their task, its doubtful that patients were processing the images the same way after multiple repetitions either due to changes in stimulus novelty and/or memory interference from previous runs. I believe it is beyond the ability of any human intracranial study to fully account for all of these different factors and so do not hold the authors up to this incredibly high standard. However, I believe this should at least be acknowledged as a possible confound in the discussion. In addition, it may be possible to rule out these alternative explanations through careful analysis of behavioral responses across runs or by comparing

firing rates amongst item and location responses using a split-halves analysis (first half of task vs second half of task).

Results: Figure 3 is a relatively unconventional way to show population results which does not allow for understanding effect sizes in single neurons or individual patients. It is also not well described in the methods and potentially obscures subsequent memory effects in non-concept cells (e.g. Line 23). I have several suggestions. First, I suggest testing and reporting subsequent memory effects at the single neuron level and adding this additional data into figure 2 (which currently doesn't provide statistical testing). Second, I would like to see results broken down further into individual subjects. For instance, I could imagine a table showing the number of cells recorded per patient or per session, the number of concept cells shown per patient, and the number of concept cells that show SMEs.

Did the authors observe conjunctive item-location neurons?

The authors report different timing amongst item and location neurons and in different brain areas based on a population analysis. To strengthen these claims, I suggest reporting cross correlation analyses between individual item and location neurons.

These recordings come from epilepsy patients. How were possible epilepsy-related confounds mitigated?

Minor comments:

What did the feedback tone indicate to the patient (line 293-294).

How often did the authors observe "several response-eliciting items (Line 347)?

Were these all images of the same concept (e.g different images of the same person)?

What information was used to determine "expected patient performance".

Reviewer #1 (Remarks to the Author):

Summary: The authors examine pre-selected neuronal activity from thousands of neurons from 15 patients (across 44 sessions) with medically refractory epilepsy during a memory task that is oriented towards locations and identities of objects that neurons were selective for. The encoding and recall portions of the task were segregated with a 'math distractor' task in which patients were instructed to count backwards from a random number for 15 seconds. The authors discovered that substantial proportions of amygdala and hippocampal neurons were selective for remembered vs. forgotten objects. This study is interesting and has cool, invaluable human neuron data and would be appropriate for publication after minor changes and added controls.

Minor critiques:

PHC isn't abbreviated in the intro until after the authors already use the full word several times.

Apart from in the Significance Statement, the full term „parahippocampal cortex“ was actually not used before the first abbreviation. The other instances were „parahippocampal location-selective neurons“ and „parahippocampal activity“. We therefore introduced the abbreviation PHC when the full term 'parahippocampal cortex' was used for the first time.

A t-test is inappropriate for the behavioral data shown for the hierarchical data in Figure 1B. A mixed effects model should be used with patient as a random factor to control for any variance contributed across subjects.

Indeed, a mixed effects linear model can account for the fact that some sessions were recorded from the same patient. We defined linear models for each of the three original subplots and replaced Figure 1B and the corresponding tests by a description of the linear mixed effects models in the Results and Methods sections:

Results:

“We used linear mixed-effects models (see Materials and Methods) to investigate the relationship between subsequent memory performance and the experimental parameters set size, reaction time and trial duration (in seconds). Our results revealed a significant effect of subsequent memory on set size ($\beta = -0.27$, $p < 10^{-16}$, subsequently forgotten trials were in larger sets) and a small but significant effect on reaction time ($\beta = 0.02$, $p = 0.01$, subsequently forgotten trials had longer reaction times), but not on trial duration ($\beta = 0.004$, $p = 0.8$).”

Methods:

“We investigated whether there was a relationship between subsequent memory performance and the three experimental parameters set size, trial duration and reaction time. To this end, we first calculated two means for each of these parameters per session, one across all subsequently remembered encoding trials and one across all subsequently forgotten encoding trials. Accordingly, subsequent memory performance was used as the predictor to model the relationship. To account for individual differences between patients and across sessions, we fitted linear mixed-effects models with random intercepts and slopes at the level of patient ID and session index (nested within patient ID). The reported estimates β in the Results section refer to the fixed slopes (i.e. average slopes across all patients).”

Important to list how many of the unit clusters were isolated units, and how many were labeled multiunit.

Good point – we added an overview table stating, among other things, the number of total clusters and single units per patient.

Patient	Session #	total units (single units)	n item cells				n location cells
			Amy	Hipp	EC	PHC	
1	1	121 (70)	0	5	4	0	
1	2	120 (67)	1	1	0	1	
1	3	57 (35)	0	1	0	0	
2	1	69 (44)	2	2	0	0	
2	2	65 (38)	4	1	0	0	
2	3	47 (31)	2	0	0	0	
2	4	32 (19)	1	0	0	0	
2	5	38 (18)	0	1	0	0	
2	6	46 (8)	0	0	0	0	
3	1	71 (50)	0	1	0	0	
3	2	83 (40)	5	3	0	0	
3	3	88 (51)	3	4	0	0	
3	4	56 (36)	3	5	0	0	
3	5	64 (46)	4	2	0	0	
4	1	86 (49)	1	1	0	0	
4	2	67 (32)	0	0	1	0	
4	3	26 (12)	0	0	0	0	
5	1	115 (64)	3	2	0	3	
5	2	108 (20)	1	2	0	3	
6	1	105 (53)	3	0	2	0	
6	2	102 (43)	6	1	4	0	
6	3	74 (27)	0	0	1	0	
6	4	66 (11)	0	1	1	0	
6	5	32 (12)	0	0	0	0	
7	1	154 (92)	0	5	7	5	
7	2	95 (40)	1	2	2	3	
7	3	90 (24)	0	1	0	1	
7	4	74 (19)	0	0	4	2	
7	5	79 (38)	0	4	2	4	
8	1	75 (35)	3	0	0	0	
8	2	76 (37)	2	2	0	0	
8	3	47 (13)	4	1	0	0	
8	4	31 (5)	3	1	0	0	
9	1	64 (25)	2	0	0	1	
10	1	152 (92)	0	2	0	1	
10	2	131 (63)	1	5	0	0	
10	3	101 (41)	0	3	1	2	
11	1	152 (87)	3	0	0	6	
11	2	133 (76)	2	5	0	2	
11	3	110 (52)	1	2	0	8	
12	1	97 (49)	8	1	1	0	
12	2	114 (61)	6	6	2	4	
12	3	85 (44)	4	5	0	1	
13	1	83 (47)	1	3	0	6	
	total	3681 (1816)	80	81	32	53	

Table 1. Units recorded per session. Overview of how many units were recorded per session, and how many were single units. The numbers of item and location cells in the different brain regions are also indicated.

It would be nice to show some sort of unit isolation or signal-to-noise ratio metrics for the units.

Thank you for this suggestion. We added this information in the Methods section, after introducing Combinato as our spike-sorting program:

“This software includes several mechanisms to automatically detect and reject artifacts: removal of spikes during extremely high firing rates, high amplitude events, overlapping spikes, and events detected concurrently on many channels. Automatically pre-sorted units were manually verified, adjusted where necessary, and classified as single units (SU), multi-units (MU), or artifacts based on spike shape and variance, signal-to-noise ratio (SNR), the inter-spike interval distribution of each cluster, and presence of a refractory period for the single units. We calculated the SNR for each single and multi-unit. It was defined as the mean spike amplitude divided by the median absolute signal. Single units (median SNR 2.85) had significantly greater SNRs than multi units (median SNR 2.08, $p < 10^{-38}$, rank-sum test).“

It looks like the EC neuron in Figure 3 has a response difference at around the same time as the PHC neurons that appears to be one of the larger effects reported in the paper. Is that difference due to outliers? Which factors did the authors correct for in their cluster based permutation tests?

As the reviewer pointed out, the EC panel in the second column shows one single neuron, which does not carry sufficient statistical power to draw any meaningful conclusions. It is rather plotted for the sake of completeness. In order to clarify the details on the cluster permutation test, we added the following in the methods section:

“The population responses in Fig. 3 were then compared using a cluster permutation test⁶⁹: first the responses during subsequently remembered and forgotten trials were compared at every time point, resulting in temporal clusters of significant differences (paired t-test $p < .05$) between the two conditions. The same was done in 10^4 iterations of label-shuffled data. Finally, the cluster sizes from the true data were ranked against the distribution of cluster sizes from the shuffled data. Only clusters whose size ranked in the top 5% were considered and marked by the grey horizontal lines in Fig. 3.“

It is unclear what group of neurons (or trials?) the authors are referring to on page 7, line 177, “...the latter group of higher firing rates...”

Thank you for pointing this out. We have expanded on this to hopefully make it clearer:

“There was a modest effect within the hippocampal location-selective neurons, where higher firing rates occurred during subsequently forgotten trials ($p = .04$). There were no effects in the other brain regions (all $p > .2$).“

In the task section of the methods, the authors refer to ‘figure 1, top row.’ Do they mean ‘Fig. 1A, top row?’

Indeed, thanks for catching this. After some adjustments Fig. 1 now only consists of the former Fig. 1A, so that in the end, the figure was adjusted to the text.

The claim that the 15 s delay constitutes a “long term memory task” may require further justification.

We agree that some supporting arguments would improve this section. We have added this statement to back up the claim:

“In this study we operationalized this association of ‘where’ and ‘what/who’, i.e. of item and location information, in the form of an associative memory task. The adaptive design increased difficulty to the point where the participants were not able to remember all information. Furthermore, every delay period was filled with a 15 second backwards counting task. This resulted in a high memory load, and prevented the rehearsal of the learned associations. Based on these features, the completion of this task requires long-term memory^{52,53}.“

where the papers referenced above are:

Jeneson, A. & Squire, L. R. Working memory, long-term memory, and medial temporal lobe function. *Learn. Mem.* **19**, 15–25 (2012).

Rose, N. S., Buchsbaum, B. R. & Craik, F. I. M. Short-term retention of a single word relies on retrieval from long-term memory when both rehearsal and refreshing are disrupted. *Mem Cogn* **42**, 689–700 (2014).

Major critiques:

Were there any differences in proportions of neurons encoding ‘what’/‘where’ in anterior/posterior hippocampal neurons?

Thank you for raising this question. We tested this with the help of paired t tests and added the statistics in the results section:

“There were two hippocampal target locations (anterior and posterior hippocampus), which were grouped together in all analyses. The fractions of responsive neurons in these two hippocampal regions did not differ within patients (item neurons: $p = .54$, location neurons: $p = .15$, paired t-test).”

Reviewer Figure 1.1. Here we plotted the fractions of item (left panel) and location (right panel) neurons in anterior hippocampus (AH) and posterior hippocampus (PH). The bars represent averages across all patients, the errorbars show standard deviations. We also added a data point for each patient, marking the fraction of responsive cells in the respective brain region across all sessions recorded with the same patient. The two hippocampal recording sites did not differ significantly in the fractions of item neurons or location neurons detected (item cells $p=.54$, location cells $p=.15$, paired t-test)

It's unsurprising that the authors find fewer location-selective neurons, as the location of the item did not need to be remembered to complete the task.

The location of the item did not have to be remembered to complete the encoding trials alone, but this information was indeed queried during the recall trials. An image was shown below an empty grid and the participants were asked to recall and tap the location that the image had been shown in during the respective encoding trial. Thus location memory was necessary to correctly complete the task.

We would suggest that the reason that fewer location-selective neurons were found lies in the pre-selection of response eliciting images with the help of the prior screening sessions. Regarding grid locations, we did not pre-select for specific locations or grid-layouts in advance, but were confined to 9 pre-defined squares in a grid.

We added the following to the introduction in order to clarify this:

“Given that we are able to pre-select items, but not locations, based on the preceding screening session (see Materials and Methods), we expected a larger number of item responses than location responses.”

There is no mention of how epileptiform discharges were removed.

As is general practice in human single-unit recordings from the medial temporal lobe, epileptiform discharges were not removed from the signals prior to spike detection and sorting (Staresina 2019 Nat Commun, Mormann 2016 Biol. Sci, Rutishauser 2010 Nature). While it is true that the frequency content of sharp epileptiform transients can partially overlap with that of action potentials and thus be falsely detected as action potentials, the vast majority of these false action potentials are easily identified as artifacts in the manual spike sorting process (see Niediek et al., 2016 and references therein). Some information on the manual spike sorting process was added. (The following section is the same as quoted above in response to the question regarding SNRs.)

“This software includes several mechanisms to automatically detect and reject artifacts: removal of spikes during extremely high firing rates, high amplitude events, overlapping spikes, and events detected concurrently on many channels. Automatically pre-sorted units were manually verified, adjusted where necessary, and classified as single units (SU), multi-units (MU), or artifacts based on spike shape and variance, signal-to-noise ratio (SNR), the inter-spike interval distribution of each cluster, and presence of a refractory period for the single units. We calculated the SNR for each single and multi-unit. It was defined as the mean spike amplitude divided by the median absolute signal. Single units (median SNR 2.85) had significantly greater SNRs than multi units (median SNR 2.08, $p < 10^{-38}$, rank-sum test).”

The paper is light on controls. What sort of controls could be carried out in order to rule out alternatives for the claims of the paper? Did they have a control brain region at all? To see if neurons' firing rates outside of the MTL are also predictive of memory performance?

Unfortunately, we do not have access to extra-mediotemporal brain regions in our recordings. There is a historic debate whether or not subdural strip electrodes are safer/less damaging than intracerebral depth electrodes. At our center, patients traditionally receive depth electrodes for temporal coverage (as it cannot be properly accessed with subdural strip electrodes) and subdural strip electrodes for frontal and extratemporal coverage. These subdural strip electrodes cannot record unit activity. Nevertheless, we report differential behavior between the amygdala, hippocampus and entorhinal cortex on the one hand and the parahippocampal cortex on the other. In this sense, each of the two groups serves as a control region for the other.

The lack of EC neuron response is contrary to much of the literature that shows EC is necessary for spatial memory. The authors should discuss this discrepancy.

Thank you for the suggestion. We have added a sentence in the introduction (bold font, introducing Reference Quasim et al., 2019)

“Spatial tuning in the form of grid cells⁹ mapping two-dimensional space on a screen has been described in the entorhinal cortex (EC) of non-human primates¹⁵ and has been linked to attention¹⁶. In humans, fMRI and MEG studies during virtual navigation have likewise provided evidence for hexagonal grid representations within the EC^{17–19}.

Another study described entorhinal cells tuned to upcoming target locations along a virtual track in humans²⁰. Nevertheless, human entorhinal neurons have been shown to generally not be involved in the processing of scenes and spatial information²¹.“

And discussed accordingly:

“The firing behavior of entorhinal neurons was of particular interest since this region lies between PHC and hippocampus along the anatomical pathway. In this experiment, we observed firing behavior in the EC to resemble that in the amygdala and hippocampus, rather than the PHC. Some previous findings point towards entorhinal involvement in spatial navigation^{20,24,48}, yet we found hardly any responses to spatial locations in this region. One possible explanation for this discrepancy is the lack of egocentric navigation required in our task. The layout on the screen is more reminiscent of a map, which is rather linked to semantic knowledge⁶². In another study, entorhinal neurons did not show the same strong preference for pictures of landscapes/scenes as the PHC⁴¹, which is in line with our findings.“

Elliot Smith (Neurosurgery) @ University of Utah

Reviewer #2 (Remarks to the Author):

In this study, the authors recorded 3681 single-units while epilepsy subjects performed an associative memory task, and demonstrate that neurons tuned to “concepts” and “location” both modulate their firing rate depending on whether the stimulus in question is remembered or not. To do so, they first identified image-selective neurons (inferring that many of these are “concept cells”) during a screening task. Then, during encoding, they identified location-selective neurons. Finally, they show that image-selective neurons in the Hipp, Amy, and EC differentiate between subsequently remembered and forgotten items, while location-selective neurons in the PHC do so. This is an interesting finding, but the key interpretations depend on analyses that require further clarification. These include (in no particular order):

1. The authors never describe how location-selective neurons are identified, simply stating that neurons “responded to at least one of the squares in the presentation grid”. What does this mean? I have to infer that the same bin-wise rank-sum test used for concept-cell identification during screening was used during stimulus presentation in a specific square during encoding. But the images are on screen for variable amounts of time – what window of time was used? Were responses between locations compared statistically (i.e. by using an ANOVA)? Was there some kind of cut-off for the # of locations that a neuron could respond to before being deemed non-selective for location?

The reviewer raises good points. Indeed, the same binwise rank-sum test was used for both item cells and location cells. We added a statement in the beginning of the referenced paragraph in the results. This also covers the concern raised in the following question, regarding the neurons’ selectivity:

“Responses to items were detected using a binwise rank-sum test ($p < .001$, see Materials and Methods). Whenever a neuron responded to one or more items, we computed the response activity for this neuron by averaging all trials containing a preferred item. Responses to grid locations were computed in the same way in that a neuron had to show a significant response to one or more of the nine locations in which an image was presented throughout the experiment. We furthermore found the responsive cells to be selective. The vast majority of these neurons responded to half or fewer of the presented images (Amygdala: 99%, Hippocampus: 96%, EC: 100%, PHC: 84%), or item locations (PHC: 81%, see below).”

The time window used is specified in the Materials and Methods section:

“To determine responsiveness, we use an established criterion based on a binwise rank-sum test (100 ms windows, 50% overlap, 0 - 1000 ms post stimulus presentation) with Simes correction for 19 bins⁴¹.”

There was no cut-off for the number of locations that a neuron could respond to before being deemed non-selective. In order to verify that we were not including a large number of cells firing to every stimulus, we considered Figure S2, a plot of the cumulative proportion of cells responding to a certain number of images of locations. Most units accumulate in the left part of the chart, which represents high selectivity or responses to only few items/locations.

2. While more straightforward, the authors do not provide any information on how many images neurons were selective for. This is important because several subsequent analyses depend only on trials including images that a neuron was selective for.

We agree, as discussed in the response to the previous question

3. The authors state that “We found significant fractions of neurons responding to locations in the amygdala, hippocampus and PHC (all $p < 10^{-15}$, binomial test)”. However, I conducted a binomial test myself in Python, where $p(\text{observing cell-counts by chance}|\alpha=0.05) = \text{scipy.stats.binom_test}(\text{cell-count}, \text{total cells}, 0.05)$, and found p-values of 0.001, 0.16, and 2×10^{-23} for amygdala, hippocampus, and PHC. Perhaps I made an error here – can the authors account for this discrepancy? This is relevant to #6, below in particular.

You can reproduce the numbers using scipy. Using your code as a starting point, this requires two adjustments: The alpha used in our response criteria is 0.001, and the test is one-sided. This would be the code to obtain our results regarding location neurons (the respective n's are stated in Figure 3, middle column):

```
#location neurons in Amy
In [7]: scipy.stats.binom_test(24, 1117, .001, alternative='greater')
Out[7]: 6.2658185393995915e-24
```

```
#location neurons in Hipp
In [8]: scipy.stats.binom_test(26, 1391, .001, alternative='greater')
Out[8]: 2.8066631713975554e-24
```

```
#location neurons in PHC
In [9]: scipy.stats.binom_test(53, 602, .001, alternative='greater')
Out[9]: 2.686826427261519e-83
```

Here are the analogous tests for item neurons:

```
#item neurons in Amy
In [5]: scipy.stats.binom_test(80, 1117, .001, alternative='greater')
Out[5]: 1.9305645839891955e-117
```

```
#item neurons in Hipp
In [6]: scipy.stats.binom_test(81, 1391, .001, alternative='greater')
Out[6]: 1.7880940017224956e-111
```

```
#item neurons in EC
In [7]: scipy.stats.binom_test(32, 571, .001, alternative='greater')
Out[7]: 1.5161835949954817e-44
```

```
#item neurons in PHC
In [8]: scipy.stats.binom_test(96, 602, .001, alternative='greater')
Out[8]: 1.4073673965634592e-175
```

To clarify this, we specified in the Results section (added text highlighted in bold font:)

“We found **significant** fractions of neurons responding to locations in the amygdala, hippocampus and PHC (all $p < 10^{-23}$, binomial test, **one-sided with $n = \text{total number of neurons per region}$, $k = \text{number of responsive neurons per region}$, $p=.001$, corresponding to the alpha level of our response criterion**)”.

and for item cells:

“Due to the preceding screening procedure, we found **highly significant** fractions of neurons responding selectively to one or more items in all recorded brain regions (**all $p < 10^{-43}$, binomial test, one-sided with $n = \text{total number of neurons per region}$, $k = \text{number of responsive neurons per region}$, $p=.001$, corresponding to the alpha level of our response criterion, refer to ‘item responses’, Fig. 2, Fig. S1A for examples**).”

4. I am not fully convinced that the two primary statistical approaches here play nice together with respect to timing. First, the authors categorize neuronal selectively by binning time (100 ms bins, 50% overlap) and looking for at least one bin that shows significantly elevated firing above baseline, corrected for multiple comparisons. If a neuron shows elevated firing in even one time-bin it is considered to be selective for item or location. However, the specific timing of elevated firing does not matter during categorization – just that neurons exceed baseline in one or multiple time-bins. There could be great variability in the specific timepoints of high firing. Yet, the cluster-permutation test used for the SME is more like an ERP, smoothing over this variability by averaging over cells and identifying specific adjacent-timepoints that distinguish remembered and forgotten trials across cells. If specific timepoints only matter for the latter analysis, and not the former, a convincing control analysis would be to categorize cells as location or item selective based on mean firing rate during stimulus presentation, and not using bin-wise categorization. This might exclude neurons who show very brief increases in firing rate, and might include neurons with more diffusely elevated firing – do the cluster permutation results survive in this case? Relying on mean firing rates could also simplify the SME calculation, resulting in a single test between recalled and forgotten trials, as in PMID: 34742943.

Thank you for these suggestions. We considered responses to be characterized by their stereotypical onset and magnitude. As you pointed out, we then used a statistical approach that averaged across many responses, smoothing out individual onsets. We actually considered it a strength in our analysis that the subsequent memory effect came through in spite of this smoothing effect. Nevertheless, we also see value in your approach of using a single large bin for calculating responsiveness. Detecting responses in this way is less conservative than our binwise approach with Simes correction. To still obtain similar numbers of responsive units, we chose a stricter alpha level for the single-bin method (rank-sum test with $\alpha=0.0001$). The pattern of emerging SMEs is quite similar (Reviewer Fig.2.2). As can be expected from the subtle differences in the smoothed instantaneous firing rates detected as statistically significant in the cluster permutation test (Fig. 3, Reviewer Fig. 2.2), an SME calculation based on a single time window was too coarse to yield comparable findings.

Reviewer Figure 2.1.

Alternative response criteria. Here we selected selective neurons according to a rank-sum test using only a single time window (1s from stimulus onset), instead of the binwise approach as in Figure 3. The alpha level was decreased to .0001 in order to obtain a criterion which is approximately as strict as the Simes-corrected binwise rank-sum test.

5. Are there any neurons that were image-selective during screening and location-selective during encoding? If so, what is their proportion? This is crucial to understanding the specificity of these results.

This is an interesting question, but not trivial to answer directly for the following reasons: the screening session had typically happened several hours before the actual grid memory paradigm. During this time, an unknown fraction of microwires move by tenths of millimeters and start recording the activity of different neurons. It would therefore not be possible to match every location neuron with its counterpart during the screening to see if it was item-selective. Both sessions were recorded and spike sorted separately, and therefore clusters would typically not match up.

The screening session is used only to find item-selective neurons whose preferred stimuli are then used in the grid memory paradigm. It would not be suitable for screening for location neurons as all images are presented at the center of the screen. The fact that we observe 7.9% item cells compared to 2.8% location cells across the MTL likely reflects the success of this screening procedure.

The question of overlap is perhaps better addressed by looking at cells within this paradigm that respond to both categories of stimuli, items and locations. Using the same response criteria, here is a modified version of Figure 3, where all cells that showed a response to both, at least 1 item and at least 1 location, were simply excluded. The effects remain for the most part, except for the EC.

Figure S4. Excluding units with both response types. These are the same responses during encoding as in Figure 3. The inclusion criteria were a significant binwise rank-sum test ($p < .001$) and responses to items or locations, but not both.

We added this as Figure S4 and refer to it in the main text (under Results / Control analyses)

“Since a small fraction of neurons were classified as both item and location neurons, we repeated the analyses after excluding these neurons and found the effects to be largely identical (Fig. S4).”

6. Judging from Figure 3, there is a very small proportion location-responsive neurons in the Amy, Hipp, and EC, yet on Line 229 the authors conclude that location-selective cells only exhibit SMEs in PHC. A more appropriate conclusion would be that location-selective cells only exist in PHC. Can the authors justify drawing conclusions about location-selective SMEs from population-level firing-rate data in Amy, Hipp, and EC, when the data come from a population small enough that it could have been observed by statistical chance? This might be relevant to image-selective cells, too (see #3).

As stated in our manuscript and in our response to point #3 of the reviewer, the fractions of location-selective cells were significant in amygdala, hippocampus and PHC (all $p < 10^{-23}$, one-sided binomial test). Location-selective neurons thus exist not only in the PHC, statistically speaking. For image-selective cells, these effects are even more significant in each of the four MTL regions (all $p < 10^{-43}$).

7. Are location cells really encoding location? How do the authors rule out that these neurons are conjunctively coding an association between items-locations, rather than location alone? One way to do so might be to measure if the same neurons fire when subjects click on the preferred locations during recall – independent of whether that recall was correct or not, the neuron should show some heightened activity in that location that is not dependent on the presence of a visual stimulus in the square.

Good point and great suggestion. We chose one second immediately before the participant's response tap on the screen to determine mean firing rates during recall. The cells included are still the same ones from our main figures: cells which responded to grid locations during encoding trials. Two average firing rates were calculated – one in response to all preferred locations ($p < .001$, binwise rank-sum test) and one in response to all other locations ($p \geq .001$, binwise rank-sum test).

We find that only in the PHC do location cells as identified during encoding consistently fire more in response to preferred locations than in response to non-preferred locations.

Figure S5.

Firing rates during retrieval. Each box plot represents cells responding to locations during encoding trials. This figure shows their average firing rates (z-scores, normalized to a baseline period of 500 ms leading up to grid onset) during retrieval in a 1 second window leading up to the response tap. Two values are computed per cell, one for response eliciting locations, one for non-response-eliciting locations (where 'response-eliciting' refers to a significant response during encoding). The mean firing rates were compared using signed-rank tests. *** $p < .001$, ** $p < .01$, * $p < .05$.

Note that even during retrieval, item cells in all four regions, but location cells only in the PHC showed significantly higher activity in response to their preferred stimuli as defined during encoding trials.

We added this figure as Supplementary Figure 5 and reference it in the main text:

“Furthermore, we verified whether the preferred stimuli of selected cells remained the same during the retrieval trials. Indeed, firing rates in response to preferred items were higher in all four recorded brain regions (all $p < .001$ for amygdala, hippocampus, EC and PHC, see Fig. S5) during retrieval. The same was true for location cells, but only in PHC ($p < .001$), supporting the idea that location cells are in fact encoding location and not merely combinations or associations. For these tests, normalized mean firing rates were computed during the 1000 ms leading up to the response tap in retrieval trials, and compared using signed-rank tests (Fig. S5).”

8. Subsequent-memory effects (SMEs) in the image-selective neurons are only ascertained in trials with image-selective stimuli. What happens in trials without image-selective stimuli? Obviously the overall firing rates may be much, much lower, but is there still a difference between remembered and forgotten items? This is important for judging the selectivity of this SME for the content that is thought to be encoded by these neurons.

This is a very interesting suggestion, and we created the respective figure. The selection of neurons and trial type is the same as in Figure 3, responsive cells ($p < .001$, binwise rank-sum test) during encoding trials. However, as suggested, the trials included were the ones containing non-response-eliciting items and locations, respectively. In order to exclude weaker, sub-threshold responses, we included images and locations with $p \geq .1$ (binwise rank-sum test). Note that the scaling on the y-axis is identical to Fig. 3 – the amplitudes are small compared to Figure 3, as you had already anticipated.

The memory effects do not emerge, except for a small, earlier effect in parahippocampal location neurons. There is an effect in the EC in the opposite direction (more activity in subsequently forgotten trials).

Reviewer Figure 2.2.

Responses to non-response-eliciting items and locations. These panels show population responses of responsive units (showing at least one response, $p < .001$, binwise rank-sum test). However, here the responses are averaged across trials containing non-response-eliciting items (left column) or locations (center column; $p \geq .1$, binwise rank-sum test). The same inclusion criterion was applied to the non-response eliciting neurons (right column), in that only trials with non-response eliciting items or locations at $p \geq .1$ were included.

9. In order to assess how the 3,681 neurons were isolated across regions, subjects, and sessions, authors should include a demographic table for the 13 subjects which includes a count of sessions-per-subject, and # of isolated neurons isolated in each region, by session.

Absolutely, good suggestion. We added a table (Table 1), with session and unit counts

Patient	Session #	total units (single units)	n item cells		n location cells	
			Amy	Hipp	EC	PHC
1	1	121 (70)	0	5	4	0
1	2	120 (67)	1	1	0	1
1	3	57 (35)	0	1	0	0
2	1	69 (44)	2	2	0	0
2	2	65 (38)	4	1	0	0
2	3	47 (31)	2	0	0	0
2	4	32 (19)	1	0	0	0
2	5	38 (18)	0	1	0	0
2	6	46 (8)	0	0	0	0
3	1	71 (50)	0	1	0	0
3	2	83 (40)	5	3	0	0
3	3	88 (51)	3	4	0	0
3	4	56 (36)	3	5	0	0
3	5	64 (46)	4	2	0	0
4	1	86 (49)	1	1	0	0
4	2	67 (32)	0	0	1	0
4	3	26 (12)	0	0	0	0
5	1	115 (64)	3	2	0	3
5	2	108 (20)	1	2	0	3
6	1	105 (53)	3	0	2	0
6	2	102 (43)	6	1	4	0
6	3	74 (27)	0	0	1	0
6	4	66 (11)	0	1	1	0
6	5	32 (12)	0	0	0	0
7	1	154 (92)	0	5	7	5
7	2	95 (40)	1	2	2	3
7	3	90 (24)	0	1	0	1
7	4	74 (19)	0	0	4	2
7	5	79 (38)	0	4	2	4
8	1	75 (35)	3	0	0	0
8	2	76 (37)	2	2	0	0
8	3	47 (13)	4	1	0	0
8	4	31 (5)	3	1	0	0
9	1	64 (25)	2	0	0	1
10	1	152 (92)	0	2	0	1
10	2	131 (63)	1	5	0	0
10	3	101 (41)	0	3	1	2
11	1	152 (87)	3	0	0	6
11	2	133 (76)	2	5	0	2
11	3	110 (52)	1	2	0	8
12	1	97 (49)	8	1	1	0
12	2	114 (61)	6	6	2	4
12	3	85 (44)	4	5	0	1
13	1	83 (47)	1	3	0	6
	total	3681 (1816)	80	81	32	53

Table 1. Units recorded per session. Overview of how many units were recorded per session, and how many were single units. The numbers of item and location cells in the different brain regions are also indicated.

10. The authors refer to their image-selective neurons as “concept cells”, despite not actually testing them for concept-specificity. I understand this follows from PMID: 28218914, but the inference being made there (and here) is that some large subset of visually-responsive neurons in these regions are concept cells. This is contradicted by Figure 4 from the same lab’s prior work (PMID: 34697305), which shows that in many of these regions (specifically PHC), many cells are only visually-responsive, and not conceptually consistent. As I point out in #6, the

proportions of image-selective cells is actually quite low outside of the PHC, and if only some subset of this already-small population is concept cells, then I hesitate to draw too many conclusions about “concept cells” from these results. I strongly suggest the authors rephrase this to “visually-selective” cells.

The number of item and location cells we find is statistically highly significant as pointed out in our responses to points #3 and #6 raised by the reviewer. However, we did not test for invariance. There are no universal criteria for defining concept cells. The distinction made in our previous publication (Bausch et al., PMID 34697305) was based on the modulation of firing rates by written questions using a rather strict statistical criterion. Further studies that focused on investigating semantic invariance are Quiroga et al. (PMID 19631538) and Rey et al. (PMID: 32142694). In Rey et al., visual invariance was high across amygdala and hippocampus (77%). In the studies by Bausch et al. and Quiroga et al., visual invariance was high across amygdala, hippocampus and EC (53.2% and 71.7%, respectively), while it was substantially lower in PHC (32.4% and 52.6%, respectively). We removed our explanation of this matter from the Introduction and Methods and instead expanded upon it in the Discussion:

Introduction:

~~Based on previous research³³ we expect a substantial fraction of visually responsive units to be concept cells, which are likely involved in memory formation⁴⁰.~~

Methods:

~~Following Kamiński et al.⁶⁹, we refer to item-selective neurons as concept cells without explicitly testing for semantic invariance.~~

Discussion:

Two independent studies have reported large fractions of visually selective neurons in the amygdala, hippocampus and EC to exhibit invariance with respect to different visual representations of the same semantic concept^{33, 54} (72% across amygdala, hippocampus and EC, and 77% across amygdala and hippocampus, respectively). A previous study from our own group required a high level of abstraction for neurons to qualify as concept cells³⁹, which, again, was the case for the majority (53%) of neurons across the same three regions. We therefore expect a majority of item neurons to qualify as concept cells.

11. The authors should cite prior evidence in humans that location-sensitive neurons show SMEs (PMID: 31902728), as well as SMEs during free recall (PMID: 34742943).

Thank you for this suggestion. We added more information to highlight the findings of PMID 31902728 (Yoo et al, 2021) in the following paragraph.

“Previous studies investigating memory encoding in the human MTL at the neuronal level have shown subsequent memory effects only at the population-code level. These studies find that the majority of memory-predictive cells show increased firing rates during encoding when information was processed that could later be recalled⁵⁵ or recognized⁵⁶. However, these effects have largely been absent in selectively responsive single neurons (6,48–50 but see 24).“

(where reference 55 is Yoo et al, 2021)

With regards to Tsitsiklis et al., we now cite this paper in a new paragraph on the entorhinal involvement in spatial navigation. However, to our understanding, there were no location-sensitive neurons showing SMEs in this work.

“The firing behavior of entorhinal neurons was of special interest since this region is closely linked to both PHC and hippocampus. In this experiment, we observed firing behavior in the EC to resemble that in the amygdala and hippocampus, rather than the PHC. Some previous findings point towards entorhinal involvement in spatial navigation^{20,24,48}, yet we found hardly any responses to spatial locations. One possible explanation for this discrepancy is the lack of egocentric navigation required in our task. The layout on the screen is more reminiscent of a map, which is rather linked to semantic knowledge⁶². In another study, entorhinal neurons did not show the same strong preference for landscapes as the PHC⁴¹, which is in line with our results.”

(where reference 48 Tsitsiklis et al, 2020)

12. The discussion of “hippocampal indexing theory” does not jive with the data as presented. I’m not positive there’s a significant # of image-selective cells in hippocampus (see point #3). I’m not sure these image-selective cells are “concept cells” (see #11). The authors do not provide a lot of support for their proposal of the PHC’s role, given the large number of image-selective cells there (with no SME) and location-selective cells (with SME).

As described in our response to the reviewer’s point #3 (and #6), we do indeed find highly significant proportions of image-selective cells in all four MTL regions. As detailed under point #10, a majority of these cells in amygdala, hippocampus and EC can be assumed to be concept neurons. Contrary to this, the PHC shows a lower degree of invariance (point #10), and stronger responses to stimuli containing spatial information (Mormann et al., 2017 PNAS). Based on these differences, we describe potential differential roles in the discussion:

“It is worth noting that location neurons in the PHC showed a subsequent memory effect, but that there was no corresponding effect in the respective item neurons in this brain region, nor in location neurons in the other three MTL regions (Fig. 3). Together with the aforementioned distinctive features of the PHC, this could indicate that the parahippocampal location neurons are not pointers, but actual neocortical representations based on a population code²¹.”

13. The discussion of the relative timing of SME effects is also not convincing given that the peak firing rate responses of both image- and location- selective cells seems to be in the same range (200-700 ms), with the SME effects in the location-selective neurons clearly aligning to the time of the button-press.

We agree that the greater latency of the SME effect in location-selective neurons is somewhat surprising. Our objective is to report statistics that represent the data as best possible and discuss them in the context of related theories from existing literature. The SME in PHC does occur immediately after the button press, but we do not see this as contradicting our discussion on the relative timing. As stated in the Results section, the effect occurs after the button press, and is thus not likely linked to any motor activity. It could however be a time window where the spatial information is consciously encoded by the participant. The task provides a suitable framework for an inner narrative that processes “what” before “where”.

This is reflected in the Discussion where we write:

“The finding that the subsequent memory effect in location responses occurred in a later time window than that for item responses could result from the way in which humans process ‘what’ and ‘where’ information. There are several linguistic models of thematic hierarchy which differ slightly depending on the phenomena they aim to explain. They rank semantic elements of sentences such as the agents, experiencers, goals, location,

instruments, etc. according to their prominence. Almost all of them rank location in the lowest category⁶³. Furthermore, there is evidence of a universal, natural order in which humans convey information when forced to use gestures instead of the spoken language they are used to. In a study where scenes with one stationary and one moving object were watched and then reproduced, the objects were acted out before the spatial movement⁶⁴. Perhaps the order of the effects we see on a neuronal level, namely item before location, reflects the architecture of internally generated narratives, where information components are processed in descending order of prominence.”

Reviewer #3 (Remarks to the Author):

Mackay and colleagues investigated the contribution of visually-responsive “concept cells” to human associative memory using microwire recordings in epilepsy patients. Following a screening session to identify images for the memory experiment, patients learned and retrieved the locations of images in a 3x3 array on a computer screen. The authors found that concept cells differentiated their firing during memory encoding and also observed neurons modulated by the location of images in the array. Other neurons did not show subsequent memory effects. The findings are interpreted as evidence that concept cells support episodic memory formation. If true, these findings would be an important and timely contribution to our understanding of the electrophysiological basis of human memory. However, the manuscript is missing key analyses and lacks sufficient description of methods for me to be wholly convinced at this point. I am nonetheless hopeful that this manuscript and dataset could be published following major revisions.

Major strengths:

The authors utilize an impressive sample size of single- and multi-unit data to assess their question.

I applaud the authors for using an adaptive and elegant memory task which allows them to test memory in patients with varying cognitive function.

Major weaknesses:

This experiment critically relies on the screening session to identify stimuli for the memory experiment yet there is very little description of the screening task. Please add a full description of the screening session.

We apologize for the lack of clarity related to our screening sessions and have added a chapter to the Methods section, describing the screening procedure in more detail.

“Screening Procedure. Each recording was preceded by a screening session in the morning of the same day in order to identify response-eliciting images. This screening session was either an object screening (OS) with a fixed set of 100 images of commonly known objects and animals described in a previous publication³², or a customized person screening (PS) with an individual set of 100-150 images of the subject’s friends and family, public figures, familiar places or objects related to their hobbies and jobs. These screenings were very similar in experimental design to the procedures described in previous publications from our own and other groups^{30,31,33,35,41,66}. Each image was shown 10 (OS) or 6 (PS) times and a simple decision task was performed after every presentation (OS: “Is the object man-made?”, PS: “Does the image contain a face?”). The repeated presentation of each image allowed for the detection of statistically significant responses to certain images. The images shown during the screening covered a large number of semantic concepts, and the stimuli selected for our main task generally depicted different objects, places or people.”

From what I can discern, only those stimuli eliciting responses in the screening session were included in the memory experiment. If true, I believe this may potentially lead to circularity in the authors conclusions because the authors did not include images that did not elicit responses as an experimental control.

It is important to note that the vast majority of the neurons we recorded and analyzed (92.8% amygdala, 94.2% hippocampus, 94.4% EC, 84.1% PHC) did not respond to any of the 4 to 8 images we selected for our memory experiment. The stimuli selected for the memory experiment typically elicited a response in only one or very few neurons. Therefore, an image that elicits a response in one a given neuron can serve as a control in most other neurons. Conversely, for a neuron responding selectively to one or few images, the remaining images can serve as controls for this neuron. We hope to have clarified this

misunderstanding with the more detailed description of our screening procedure listed under the previous point.

Images were re-used across runs in a session. While the authors audibly included a “confirmation tap” in their task, its doubtful that patients were processing the images the same way after multiple repetitions either due to changes in stimulus novelty and/or memory interference from previous runs. I believe it is beyond the ability of any human intracranial study to fully account for all of these different factors and so do not hold the authors up to this incredibly high standard. However, I believe this should at least be acknowledged as a possible confound in the discussion. In addition, it may be possible to rule out these alternative explanations through careful analysis of behavioral responses across runs or by comparing firing rates amongst item and location responses using a split-halves analysis (first half of task vs second half of task).

Thank you for this suggestion. We did a split-halves analysis to address this concern. To this end, we recreated Figure 3, but for each response we only included the first or second half of trials. When an item was presented an odd number of times, the middle trial was excluded entirely.

Reviewer Fig. 3.1.a
Split-halves analysis, 1st half

Reviewer Fig. 3.1.b

Split-halves analysis, 2nd half.

Figures 3.1.a and 3.1.b are similar to Figure 3, but for each response, they only include the first half (3.1.a) or second half (3.1.b) of all presentation trials.

We find the emerging effects to be quite stable between the first and second half of trials and the original Figure 3.

We have therefore added a paragraph to the Results section:

“Given that images were shown repeatedly across trials, we tested for effects of adaptation or memory interference from previous trials by performing a split-half analysis (first half of trials vs second half of trials). Both halves showed quantitatively similar results to those shown in Fig. 3 (data not shown here).”

Results: Figure 3 is a relatively unconventional way to show population results which does not allow for understanding effect sizes in single neurons or individual patients. It is also not well described in the methods and potentially obscures subsequent memory effects in non-concept cells (e.g. Line 23). I have several suggestions. First, I suggest testing and reporting subsequent memory effects at the single neuron level and adding this additional data into figure 2 (which currently doesn't provide statistical testing). Second, I would like to see results broken down further into individual subjects. For instance, I could imagine a table showing the number of cells recorded per patient or per session, the number of concept cells shown per patient, and the number of concept cells that show SMEs.

We updated figure 2 by statistically testing for SMEs for each of the four displayed neurons. This information has now been added to the figure caption:

“Subsequent memory effects per neuron were statistically assessed using a Wilcoxon rank-sum test for the time window of 0 to 1500 ms. Statistically significant effects were found for the two item neurons (top, $p = 0.008$; bottom, $p = 0.02$), but not for the two location neurons (both $p > 0.1$).”

Note that since we are explicitly reporting SMEs here and the previous top left unit in figure 2 (located in the amygdala) did not show this effect, we exchanged this example for a neuron with higher firing rates and a significant SME.

Figure 2. Examples of item and location-specific responses. (A) Selective responses by single neurons (top: amygdala, bottom: hippocampus), separated based on correct vs. incorrect subsequent retrieval. Solid lines (lower panels): response to the preferred item. Dashed lines: average response to all non-preferred items (cf. Fig. S1A). (B) Responses of single neurons in the PHC to spatial locations within the presentation grid. Solid lines (lower panels): response to the preferred item locations, which in the lower example includes the entire bottom row of the grid. Dashed lines: average response to all non-preferred locations (cf. Fig. S1B) Subsequent memory effects per neuron were statistically assessed using a Wilcoxon rank-sum test for the time window of 0 to 1500 ms. Statistically significant effects were found for the two item neurons (top, $p = 0.008$; bottom, $p = 0.02$), but not for the two location neurons (both $p > 0.1$).

We also added a table showing various cell counts such as single units, item and location cells, per session.

Patient	Session #	total units (single units)	n item cells				n location cells
			Amy	Hipp	EC	PHC	
1	1	121 (70)	0	5	4	0	
1	2	120 (67)	1	1	0	1	
1	3	57 (35)	0	1	0	0	
2	1	69 (44)	2	2	0	0	
2	2	65 (38)	4	1	0	0	
2	3	47 (31)	2	0	0	0	
2	4	32 (19)	1	0	0	0	
2	5	38 (18)	0	1	0	0	
2	6	46 (8)	0	0	0	0	
3	1	71 (50)	0	1	0	0	
3	2	83 (40)	5	3	0	0	
3	3	88 (51)	3	4	0	0	
3	4	56 (36)	3	5	0	0	
3	5	64 (46)	4	2	0	0	
4	1	86 (49)	1	1	0	0	
4	2	67 (32)	0	0	1	0	
4	3	26 (12)	0	0	0	0	
5	1	115 (64)	3	2	0	3	
5	2	108 (20)	1	2	0	3	
6	1	105 (53)	3	0	2	0	
6	2	102 (43)	6	1	4	0	
6	3	74 (27)	0	0	1	0	
6	4	66 (11)	0	1	1	0	
6	5	32 (12)	0	0	0	0	
7	1	154 (92)	0	5	7	5	
7	2	95 (40)	1	2	2	3	
7	3	90 (24)	0	1	0	1	
7	4	74 (19)	0	0	4	2	
7	5	79 (38)	0	4	2	4	
8	1	75 (35)	3	0	0	0	
8	2	76 (37)	2	2	0	0	
8	3	47 (13)	4	1	0	0	
8	4	31 (5)	3	1	0	0	
9	1	64 (25)	2	0	0	1	
10	1	152 (92)	0	2	0	1	
10	2	131 (63)	1	5	0	0	
10	3	101 (41)	0	3	1	2	
11	1	152 (87)	3	0	0	6	
11	2	133 (76)	2	5	0	2	
11	3	110 (52)	1	2	0	8	
12	1	97 (49)	8	1	1	0	
12	2	114 (61)	6	6	2	4	
12	3	85 (44)	4	5	0	1	
13	1	83 (47)	1	3	0	6	
total		3681 (1816)	80	81	32	53	

Table 1. Units recorded per session. Overview of how many units were recorded per session, and how many were single units. The numbers of item and location cells in the different brain regions are also indicated.

We also followed the reviewer's suggestion and calculated one rank-sum test with an alpha level of .05 per responsive cell, comparing subsequently remembered and subsequently forgotten trials in the same time windows of 0 to 1500ms post stimulus onset as in Fig. 2.

The fractions of single neurons showing a statistically significant SME were rather small and reached statistical significance (one-sided binomial test with n = number of responsive neurons per region, k = number of responsive neurons per region showing SME, p = .05, corresponding to the

alpha level of our rank-sum test for SME) for item neurons in the amygdala ($p = 0.007$), but not in hippocampus or EC (both $p > 0.5$). Likewise, the fraction of location neurons in PHC showing significant SMEs did not reach statistical significance ($p = 0.3$).

It is for this very reason that we performed a second-level analysis across the group of item and location cells, respectively, in order to assess their population effect.

Did the authors observe conjunctive item-location neurons?

This is a very good question. We were not able to test for responses to every individual item-location combination, as there were simply too many combinations ($9 * n_{stimuli}$), and insufficient trial counts.

We did, however, find neurons with responses to items AND locations. In order to rule out that the subsequent memory effects rely on these neurons alone, we excluded them from the analysis shown in Figure 3. The effects largely remain.

Figure S4

Excluding units with both response types. These are the same responses during encoding as in Figure 3. The inclusion criteria were a significant binwise rank-sum test ($p < .001$) and responses to items or locations, but not to both.

The authors report different timing amongst item and location neurons and in different brain areas based on a population analysis. To strengthen these claims, I suggest reporting cross correlation analyses between individual item and location neurons.

We computed cross correlations between pairs consisting of one parahippocampal location neuron and one item neuron in either amygdala, hippocampus or EC. Both neurons naturally had to be

measured during the same recording session and had to be located within the same hemisphere. Thus, the groups of pairs were:

pair type 1: Amygdala (item cells)	& PHC (location cells)	n = 31 pairs
pair type 2: Hippocampus (item cells)	& PHC (location cells)	n = 67 pairs
pair type 3: EC (item cells)	& PHC (location cells)	n = 44 pairs

For each pair, we computed asymmetry scores as follows:

$$(nPos - nNeg) / (nPos + nNeg)$$

where nPos is the number of positive lags and nNeg is the number of negative lags in the cross correlogram.

We considered spikes in the time windows marked by the memory effects as seen in Figure 3 (grey horizontal bars). These were 230 – 1444ms (pair type 1), 530 – 1444ms (pair type 2) and 490-1444ms (pair type 3), and maximum absolute lags of 100ms. We did not find the asymmetry scores to significantly deviate from zero in any of the groups of pairs (pair type 1: $p=.54$, pair type 2: $p=.17$, pair type 3: $p=.26$).

The microwires we record these units with lie several millimeters apart. We assume that the pairs of neurons we were able to record here were simply not (or not directly) connected, and thus did not show consistent orders in their firing.

These recordings come from epilepsy patients. How were possible epilepsy-related confounds mitigated?

With regard to the physiological recordings, there were several mechanisms in place in order to remove artifacts. In order to clarify this, we added the following section immediately after introducing our spike sorting software Combinato.

“This software includes several mechanisms to automatically detect and reject artifacts: removal of spikes during extremely high firing rates, high amplitude events, overlapping spikes, and events detected concurrently on many channels. Automatically pre-sorted units were manually verified, adjusted where necessary, and classified as single units (SU), multi-units (MU), or artifacts based on spike shape and variance, signal-to-noise ratio (SNR), the inter-spike interval distribution of each cluster, and presence of a refractory period for the single units. We calculated the SNR for each single and multi-unit. It was defined as the mean spike amplitude divided by the median absolute signal. Single units (median SNR 2.85) had significantly greater SNRs than multi units (median SNR 2.08, $p < 10^{-38}$, rank-sum test).“

Based on previous research, we do not expect epilepsy-related firing behavior to have a significant impact on our recordings. We are specifically referring to Lee et al. (2021, „Single-neuron correlate of epilepsy-related cognitive deficits in visual recognition memory in right mesial temporal lobe“). This study looked at visually selective cells and memory-related cells in the MTL of epilepsy patients. They found that the effect of being located in a SOZ on the neurons' tuning was largely confined to a specific subgroup during specific behavior, namely memory-selective cells during high-confidence retrieval. Furthermore we would argue that even if some of the selective cells we recorded were showing abnormal behavior related to epilepsy, this would potentially only obscure the effects we found, not artificially produce them.

We added another paragraph to the Methods/Task section:

“We did not expect epilepsy-related neuronal firing to substantially affect our results. As shown by Lee et al.⁶⁷, such interference should be minimal and should only affect small, specific sub-populations during recall.”

Minor comments:

What did the feedback tone indicate to the patient (line 293-294).

Thank you for drawing our attention to this. We added a more detailed and clearer description in the text (added text in bold font):

“The participant was asked to confirm every image location by tapping it within the presentation time window (1.5 – 3 s). **Whenever the correct square was tapped, a green frame appeared along its outline for the remainder of the presentation duration.** Trials with off-target or missing confirmation tap were considered invalid and were excluded from the analyses. These trials also triggered an immediate **dissonant** feedback sound and, in case of a misplaced tap, a red frame **around the tapped, empty square.**“

How often did the authors observe “several response-eliciting items” (Line 347)?

This information is indirectly contained in Figure S2. A line graph representing a very selective population reaches 1 on the y axis very quickly, meaning that almost all cells responded to 0 or very few images. You can see that at $x = 0.5$, all brain regions except PHC are almost at 1. Since this is a cumulative plot, it means that hardly any neurons showed responses to more than half of the images. This figure refers to the entire population of recorded cells.

We added the following statistics regarding the responsive cells in the Results section:

„We furthermore found the responsive cells to be selective. A high fraction responded to half or fewer of the presented images (Amygdala: 99%, Hippocampus: 96%, EC: 100%, PHC: 84%), or grid locations (PHC: 81%).“

Were these all images of the same concept (e.g different images of the same person)?

Thank you for bringing this up. To briefly answer your question: no, the images generally showed very different objects or individuals. In order to make the process of image selection clearer, we have added a section on the screening procedure in the Methods:

“**Screening Procedure.** Each recording was preceded by a screening session in the morning of the same day in order to identify response-eliciting images. This screening session was either an object screening (OS) with a fixed set of 100 images of commonly known objects and animals described in a previous publication³², or a customized person screening (PS) with an individual set of 100-150 images of the subject’s friends and family, public figures, familiar places or objects related to their hobbies and jobs. These screenings were very similar in experimental design to the procedures described in previous publications from our own and other groups^{30,31,33,35,41,66}. Each image was shown 10 (OS) or 6 (PS) times and a simple decision task was performed after every presentation (OS: “Is the object man-made?”, PS: “Does the image contain a face?”). The repeated presentation of each image allowed for the detection of statistically significant responses to certain images. The images shown during the screening covered a large number of semantic concepts, and the stimuli selected for our main task generally depicted different objects, places or people.”

What information was used to determine “expected patient performance”.

There was no hard criterion at our disposal, but it was obvious from the beginning that a patient who would only be able to retain small numbers of items at a time (e.g. on average 2-3), would not be able to accumulate sufficient trials within the 35-minute time frame of the task if we included 8 different images. On the other hand, a high-performing individual needed a larger stimulus pool of

up to 8 images so that sufficiently large sets could be presented. Things we took into consideration when putting together the stimulus pool were the neuropsychological assessment from before their admission, but also the general impression that patients' cognition made after the implantation.

REVIEWER COMMENTS

Reviewer #1 (Remarks to the Author):

The authors have adequately addressed all of my concerns. This is a great contribution to the human single neuron, and neuroscience literature.

Reviewer #2 (Remarks to the Author):

I commend the authors for addressing many of my comments, and find the overall paper strengthened as a result. However, a couple of key issues remain.

1) I am not sure I follow the authors' reasoning pertaining to their binomial tests. In the code snippet provided, why set the probability of detecting a significant neuron at 0.1%? Is there really only a 1/1000 chance of observing a location-selective neuron in the hippocampus? This is extremely low - setting such an extremely lenient probability of observation lowers the bar for a significant fraction of location- or item- tuned neurons dramatically, and this affects the interpretations throughout the paper. Even observing a paltry 4/1117 location neurons in the Amygdala, for example, would yield a p-value < 0.05:

```
scipy.stats.binom_test(4, 1117, .001, alternative='greater') -> 0.02694
```

I would be hard-pressed to argue that 4 neurons exhibiting an effect out of 1117 were a significantly greater proportion than expected by chance.

Most authors in the field set this proportion at 5% (PMID: 26139375, 34265253). Even the senior author's own prior work have set the expected proportion of significant neurons at 5% (PMID: 29206940). Did the authors intend to set a far more lenient statistical threshold than prior studies? If so, I think a sufficient justification would need to be provided for deviating so dramatically from the typical implementation of such a test. If I am missing something, please clarify.

Alternatively, the authors could empirically determine the number of neurons expected to be significant by chance by performing permutation testing with scrambled data (PMID: 35260859, 28218914).

The result of this analysis is important to prior comments #1, #6, #10, and #12, and thus clarifying it would be helpful to accurately assessing the authors' downstream claims.

2. I don't think Reviewer Figure 2.1 really addresses my point as the authors adjusted the alpha to match cell counts. Perhaps a more informative request would be this - can the authors provide a histogram of all the timepoints deemed significant for the individual item and location neurons? It is important to compare this histogram to the time deemed significant by the group-level cluster permutation tests.

3. Could the authors provide more interpretation of the interesting result in Figure S5? What does it imply (if anything) that only PHC-localized location cells continue to exhibit elevated firing in the preferred location during recall? Is the visual stimulus in the square partially responsible for "location cell" activity in the other regions?

Reviewer #3 (Remarks to the Author):

Thank you for your thorough response to my comments. The authors have now sufficiently addressed my concerns and I believe the manuscript is now suitable for publication. Nice work!

Reviewer #1 (Remarks to the Author):

The authors have adequately addressed all of my concerns. This is a great contribution to the human single neuron, and neuroscience literature.

We thank the reviewer for their insightful comments and the positive assessment of our work.

Reviewer #2 (Remarks to the Author):

I commend the authors for addressing many of my comments, and find the overall paper strengthened as a result. However, a couple of key issues remain.

1) I am not sure I follow the authors' reasoning pertaining to their binomial tests. In the code snippet provided, why set the probability of detecting a significant neuron at 0.1%? Is there really only a 1/1000 chance of observing a location-selective neuron in the hippocampus? This is extremely low - setting such an extremely lenient probability of observation lowers the bar for a significant fraction of location- or item- tuned neurons dramatically, and this affects the interpretations throughout the paper. Even observing a paltry 4/1117 location neurons in the Amygdala, for example, would yield a p-value < 0.05 :

```
scipy.stats.binom_test(4, 1117, .001, alternative='greater') -> 0.02694
```

I would be hard-pressed to argue that 4 neurons exhibiting an effect out of 1117 were a significantly greater proportion than expected by chance.

Most authors in the field set this proportion at 5% (PMID: 26139375, 34265253). Even the senior author's own prior work have set the expected proportion of significant neurons at 5% (PMID: 29206940). Did the authors intend to set a far more lenient statistical threshold than prior studies? If so, I think a sufficient justification would need to be provided for deviating so dramatically from the typical implementation of such a test. If I am missing something, please clarify.

Thank you for raising this important point. In our binomial test, we set $p=.001$ to match the alpha criterion we use in our response detection (binwise rank-sum test where at least one bin had to fire above baseline with $p < .001$ after Simes correction, as in PMID 18768680, 21874014, 31158216 and 28943091). For our analysis, the binomial test is directly linked to our response detection in that it tests whether the number of detected responses is significantly higher than expected by chance. Concept cell responses are generally very strong, sparse and selective, such that a rather conservative threshold (.001) is appropriate to avoid false positives.

In the papers referenced by the reviewer, either a different response criterion was used or the binomial test was linked to a different type of analysis:

- PMID 26139375: in Ison et al., the binomial test with $p=.05$ is tied to a (more liberal) response criterion using a Wilcoxon rank-sum test with $p=.05$
- PMID 34265253: Kunz et al. generally use ANOVAs to detect different types of cells (such as anchor cells), then run a binomial test to check for significant proportions, and use a threshold of $p < .05$ in almost all tests.
- PMID 29206940: Mormann et al. report "a binomial test using an expected false positive probability of $P = 0.05$, identical to the alpha-level of the Spearman correlations"

That being said, we do realize that the choice of p in the binomial test reflects an *analytically derived size* which may not necessarily match its *empirical size*. We therefore have followed the reviewer's next suggestion (see below).

Alternatively, the authors could empirically determine the number of neurons expected to be significant by chance by performing permutation testing with scrambled data (PMID: 35260859, 28218914).

The result of this analysis is important to prior comments #1, #6, #10, and #12, and thus clarifying it would be helpful to accurately assessing the authors' downstream claims.

We are very grateful for this suggestion. Assessing the empirical rather than the nominal size (i.e. the probability of falsely rejecting the null hypothesis if it is true) has changed the findings as follows: We now find significant proportions of *location neurons* only in the PHC, but no longer in the amygdala and hippocampus.

We now describe the label-shuffling permutation test in the Methods and Results sections and have added a supplementary figure (Figure S6) showing the label-shuffled distributions.

Label-shuffling is now added in the Results section:

in the context of item neurons (new text in bold)

“In addition to the binomial test, we calculated the empirical size (i.e., the probability of falsely rejecting the null hypothesis if it is true) in each measured brain region. To this end, we compared the fraction of responsive items to 10000 iterations of label-shuffled data, and found empirical sizes of $\alpha < 10^{-4}$ in all measured brain regions (Fig. S6). Thus, for item neurons the binomial test (nominal size) and label-shuffling test (empirical size) produced consistent results.”

in the context of location neurons

“Using a binomial test, we found significant fractions of neurons responding to locations in the amygdala, hippocampus and PHC (all $p < 10^{-23}$, one-sided with n = total number of neurons per region, k = number of responsive neurons per region, p = .001, corresponding to the **size (i.e., alpha level) of our response criterion). **It is important to note, however, that the nominal size (significance level) might not always align with the empirical size of the test. To specifically test whether more than .001 of cells could be expected to be responsive by chance in our data, we used a permutation test to compare the measured fractions of location neurons to 10000 iterations of label-shuffled data. We found a significant percentage of location neurons in the PHC ($p=0.0027$), but not in the amygdala ($p=0.23$) and hippocampus ($p=0.81$, Fig. S6). We then statistically compared the proportion of location cells found in PHC to that in all other regions.”****

as well as the Methods section

“The number of responsive neurons was then tested against chance level for each brain region, using two different approaches. One was a parametric approach, a binomial test where the occurrence rate p was set to $p = .001$, the same as the alpha level in the response criterion described above. The other was a permutation-based approach where either item or location labels were shuffled 10000 times, resulting in a distribution of 10000 proportions (i.e., surrogate realizations). The p-value was calculated as the fraction of label-shuffled data points that were higher than the measured data.”

Supplemental Figure 6: Empirical sizes of the permutation tests. The histograms show fractions of neurons producing significant responses (see Materials and Methods) to specific items (top) or locations (bottom) after label shuffling. Blue bars represent distributions of 10000 iterations, red lines represent fractions for the actual measured data.

Evidently these additional results, more specifically than our original analyses, highlight the role of the PHC with regard to location. Once again we thank the reviewer for their diligent efforts. Your comprehensive understanding and active involvement have enhanced the quality of our work.

2. I don't think Reviewer Figure 2.1 really addresses my point as the authors adjusted the alpha to match cell counts. Perhaps a more informative request would be this - can the authors provide a histogram of all the time points deemed significant for the individual item and location neurons? It is important to compare this histogram to the time deemed significant by the group-level cluster permutation tests.

Thank you for following up on this. We addressed your question in two ways:

First, the time points deemed significant for the individual item and location neurons *in terms of the responses*. Here we show histograms of the time bins where the binwise rank-sum test was significant (before Simes correction). The counts of item and location neurons with a significantly elevated firing rate at a given time bin are compared to distributions of 10k counts of label-shuffled data. We added a second panel for each brain region (Reviewer Figure 1A and B, lower panels) showing the time points at which the measured data fell into the top 5% of the label-shuffled data.

Reviewer Figure 1A shows that the latencies at which significant item responses were detected on a single neuron basis overlap with the time windows in which subsequent memory effects are detected in the amygdala, hippocampus and EC (Figure 3).

Reviewer Figure 1B shows the same for location responses in the PHC. The time bins between 1000 and 1500 ms post stimulus onset show subsequent memory effects in high numbers of neurons as compared to the shuffled data, with one bin scoring above the 95th percentile.

A: Item responses

B: Location responses

Reviewer Figure 1: Item (A) and location (B) responses in measured vs. shuffled data.
Top row: measured data (grey) and medians of 10k iterations of shuffled data (red). For each time bin, the absolute number of neurons is shown for which the time bin produced a significant response before Simes correction (binwise ranksum test). There is no red histogram for the EC since very few responsive cells were present in label-shuffled data, and their median firing rate was zero. Note that the time bins used for these calculations overlap by 50%, but the bars in this visualization do not. Since the top panels do not show the variance within the shuffled data, we calculated the percentile that the measured data falls into at each time bin. This data is shown in the bottom row. The red line marks the 95th percentile.

Second, the time points deemed significant for individual item and location neurons in terms of a *subsequent memory test*. Here, the histograms include the same overlapping time bins as in the analysis above. For each unit, we only considered time bins in which the unit produced a significant response according to the binwise rank-sum test before Simes correction. (These are the same bins that we visualized in the histograms above.) We then calculated whether there was a subsequent memory effect in each of those bins (one-sided rank-sum test, $p < .05$)

A: Item responses

B: Location responses

Reviewer Figure 2: Memory effects in item (A) and location (B) responses in measured vs. shuffled data. Top row: measured data (grey) and medians of 10k iterations of shuffled data (red). The counts refer to neurons with a significantly elevated firing rate at each bin according to the response criterion before Simes-correction (binwise ranksum test, $p < .001$) and a significantly higher firing rate during subsequently remembered than forgotten trials (rank-sum test, one-sided, $p < .05$). Note that the time bins used for these calculations overlap by 50%, but the bars in this visualization do not. Since the top panels do not show the variance within the shuffled data, we calculated the percentile that the measured data falls into at each time bin. This data is shown in the bottom row. The red line marks the 95th percentile.

As before, we were not able to detect subsequent memory effects on a single-neuron basis, most likely owing to limited statistical power per neuron.

3. Could the authors provide more interpretation of the interesting result in Figure S5? What does it imply (if anything) that only PHC-localized location cells continue to exhibit elevated firing in the preferred location during recall? Is the visual stimulus in the square partially responsible for “location cell” activity in the other regions?

Thank you for following up on this. The “location cell” activity outside of the PHC has lost its statistical significance after performing the label-shuffling test the reviewer suggested. We added a reference to these findings in the discussion:

“Since the fractions of location cells in the amygdala and hippocampus were not statistically significant in the label-shuffling permutation test, any response activity to a specific grid location in these regions (Fig. 3) is likely an epiphenomenon of response activity to the visual stimuli. We only found a significantly large population of location cells in the PHC, which was also the only brain region to respond more strongly to the same preferred locations during retrieval as during encoding (Fig. S5).”

Reviewer #3 (Remarks to the Author):

Thank you for your thorough response to my comments. The authors have now sufficiently addressed my concerns and I believe the manuscript is now suitable for publication. Nice work!

We thank the reviewer for their insightful comments and the positive assessment of our work.

REVIEWERS' COMMENTS

Reviewer #2 (Remarks to the Author):

The authors have rigorously addressed my concerns, and I believe this work will be a valuable addition to the field.

- Salman Qasim